# Peri-LN: Revisiting Normalization Layer in the Transformer Architecture

**Jeonghoon Kim** [1 2]  **Byeongchan Lee** [2]  **Cheonbok Park** [1 2]  **Yeontaek Oh** [1]  **Beomjun Kim** [2]  **Taehwan Yoo** [1]
**Seongjin Shin** [1]  **Dongyoon Han** [3]  **Jinwoo Shin** [† 2]  **Kang Min Yoo** [† 1]

## Abstract

Selecting a layer normalization (LN) strategy that stabilizes training and speeds convergence in Transformers remains difficult, even for today's large language models (LLM). We present a comprehensive analytical foundation for understanding how different LN strategies influence training dynamics in large-scale Transformers. Until recently, Pre-LN and Post-LN have long dominated practices despite their limitations in large-scale training. However, several open-source models have recently begun silently adopting a third strategy without much explanation. This strategy places normalization layer *peripherally* around sublayers, a design we term *Peri-LN*. While Peri-LN has demonstrated promising performance, its precise mechanisms and benefits remain almost unexplored. Our in-depth analysis delineates the distinct behaviors of LN strategies, showing how each placement shapes activation variance and gradient propagation. To validate our theoretical insight, we conduct extensive experiments on Transformers up to 3.2B parameters, showing that Peri-LN consistently achieves more balanced variance growth, steadier gradient flow, and convergence stability. Our results suggest that Peri-LN warrants broader consideration for large-scale Transformer architectures, providing renewed insights into the optimal placement of LN.

## 1. Introduction

Building on a rapidly expanding lineage of Transformer-based large language models, open-source models have

[†]Equal correspondence. [1]NAVER Cloud [2]Korea Advanced Institute of Science and Technology (KAIST) [3]NAVER AI Lab. Correspondence to: Jeonghoon Kim <jeonghoon.samuel@gmail.com>, Jinwoo Shin <jinwoos@kaist.ac.kr>, Kang Min Yoo <kangmin.yoo@navercorp.com>.

*Proceedings of the 42$^{nd}$ International Conference on Machine Learning*, Vancouver, Canada. PMLR 267, 2025. Copyright 2025 by the author(s).

shown remarkable impact (Hoffmann et al., 2022; Guo et al., 2025; Yoo et al., 2024). As the demand for larger and more powerful models grows, various training stabilization techniques have been introduced (Yang et al., 2022; Zhai et al., 2023; Loshchilov et al., 2024). Among these, the choice of where and how to apply layer normalization (LN: LayerNorm or RMSNorm; Ba et al., 2016; Zhang & Sennrich, 2019) critically influences model convergence (Xiong et al., 2020; Kedia et al., 2024; Wortsman et al., 2024). However, their immense computational requirements have restricted deeper exploration of the underlying Transformer structure. Are we truly employing the optimal LN placement? In practice, fully revealing the results of massive resource investments can be challenging (Rivière et al., 2024). Despite its importance, there is still no consensus on a single best LN placement strategy.

Two prominent LN placements have been widely explored. Post-LN (Vaswani et al., 2017) normalizes the hidden state after adding the sub-layer output to the residual stream (that is, $\text{Norm}(x + \text{Module}(x))$ where $x$ is input hidden state. $\text{Norm}$ is LN). This helps constrain the variance of hidden states but may inadvertently weaken gradient signals, particularly in deeper models (Kedia et al., 2024). Pre-LN (Dubey et al., 2024), by contrast, normalizes before passing the hidden state to the sub-layer (that is, $x + \text{Module}(\text{Norm}(x))$). While this can enhance gradient propagation, it also admits so-called "massive activations," where hidden states grow exponentially across layers (Sun et al., 2024).

Previous studies on deep convolutional neural networks (CNNs) have analyzed the impact of batch normalization on variance changes during the initialization stage of ResNet architectures, demonstrating its relationship to model performance (De & Smith, 2020). They noted that, in models without normalization, hidden activation growth *at initialization* can be exponential, leading to poor performance and stability. In contrast, in pre-normalized CNNs, the variance of hidden activations was shown to increase linearly as model depth grows. In the same vein, Kedia et al. (2024) reported that, for Transformer architectures as well, the variance in the forward propagation of Transformer-based language models *at initialization* increases linearly with depth. However, in the context of Transformer architectures, we observed that this variance growth at initialization does

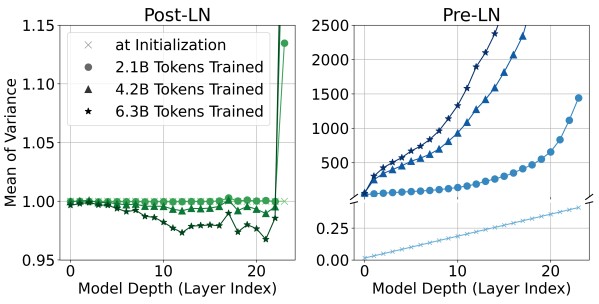

*Figure 1.* Illustration of hidden-state variance across different model depths and training iterations. From initialization through training on 6.3 billion tokens, we observe the growth in hidden-state variance for both Pre-LN and Post-LN architectures. The analysis is based on a 1.5B-parameter model. Detailed settings and additional results are provided in Section 5.1.

not persist as training progresses as shown in Figure 1. Sections 3, 4, 5, and 6 provide a more detailed discussion of these hidden-state growth patterns.

Beyond these two common strategies, Post-LN and Pre-LN, a third LN placement has quietly emerged in large-scale open-source models: applying LN around the sub-layer, i.e., on both its input and output. Although recent open-source models (Team et al., 2025; Rivière et al., 2024; OLMo et al., 2024) have quietly adopted such designs and demonstrated promising performance on a large scale, these efforts often appeared isolated, lacking a conceptual unifying framework or a thorough investigation into their benefits. In this paper, we coin the term *Peri-LN*[1] to unify these scattered approaches and highlight an underexplored avenue for stabilizing large-scale Transformer training. By dissecting the forward- and backward-pass dynamics of each LN strategy, we clarify how, when, and why they differ, interpreting these distinctions through the lens of training stability.

Accordingly, this paper revisits LN placement in Transformers from both analytical and empirical perspectives. In particular, we:

1. Present an in-depth analysis of Post-LN and Pre-LN in large-scale Transformers, examining how variance and gradient properties evolve *beyond initialization*.

2. Investigate Peri-LN to understand how normalizing both the inputs and outputs of each module moderates hidden-state behavior during forward and backward propagation, providing a systematic perspective on this underexplored alternative.

3. Provide quantitative evidence on how large activation influences training stability, benchmark performance, and model behaviors.

---

[1]*Peri-* (Prefix) means "around," reflecting that LN encapsulates the entire sub-layer. (e.g. peripherally)

## 2. Background and Motivation

The analysis of activation variance at model initialization has long been central to understanding normalization layers and enhancing stability in convolutional neural networks (CNNs) (De & Smith, 2020; He et al., 2016; Brock et al., 2021a). Specifically, De & Smith (2020) showed that batch normalization in residual blocks can bias networks toward the identity function, thereby stabilizing gradients and improving overall training dynamics.

Similar investigations have emerged for Transformer architectures, examining how variance propagates and how gradients behave in both post-layer normalization (Post-LN) and pre-layer normalization (Pre-LN) configurations (Xiong et al., 2020; Kedia et al., 2024; Wortsman et al., 2024). Early work comparing Post- and Pre-LN primarily focused only on the initialization stage. Xiong et al. (2020) observed that Pre-LN architectures tend to exhibit more stable gradients, but can still encounter issues such as gradient spikes and divergence, especially in deeper models or large-scale pre-training scenarios (Zhai et al., 2023; Wortsman et al., 2024; Fishman et al., 2024; Chung et al., 2024).

Among these challenges, the phenomenon of "massive activations" has attracted attention (Dettmers et al., 2022; Yu et al., 2024; Fishman et al., 2024). Notably, Sun et al. (2024) identified that in Pre-LN architectures, large spikes in activation magnitude can persist across layers due to residual connections. These massive activations act as fixed biases, potentially narrowing the model's focus to certain tokens and may influence generalization. However, the underlying mechanisms behind these large values, and their exact impact on the training process, remain not yet well understood.

Analytical work has provided theoretical frameworks to explain phenomena like gradient explosion and vanishing in Transformers. For instance, Kedia et al. (2024) introduced a signal propagation theory that details how activation variance and gradient instability can evolve with depth, identifying critical factors that impair stability and performance. Recent studies have discussed how Pre-LN architectures can allow large values from Attention or MLP modules to flow unimpeded through residual connections (Csordás et al., 2024; Fishman et al., 2024; Zhai et al., 2023; Wortsman et al., 2024), but the precise impact of this behavior on large-scale training remains insufficiently explored.

These observations underscore the ongoing need to clarify how activation dynamics and normalization strategies interact, especially in large-scale training.

**In response,** this work aims to deepen our understanding of how normalization strategies influence Transformer training, with particular attention to the emergence of large activations and their implications for stability and performance.

# 3. Normalization Strategies

In this section, we discuss how different placements of layer normalization (LN [2]) in Transformer architecture affect both training stability and the statistics of hidden states (activations [3]).

## 3.1. Post- & Pre-Normalization in Transformers

**Post-LN.** The Post-Layer Normalization (Post-LN) (Vaswani et al., 2017) scheme, normalization is applied *after* summing the module's output and residual input:

$$y_l = \text{Norm}\big(x_l + \text{Module}(x_l)\big), \qquad (1)$$

where $x_l$ is the input hidden state of $l$-th layer, $y_l$ is the output hidden state of $l$-th layer, and Module denotes Attention or Multi-Layer Perceptron (MLP) module in the Transformer sub-layer. Norm denotes normalization layers such as RMSNorm or LayerNorm. It is known that by stabilizing the activation variance at a constant scale, Post-LN prevents activations from growing. However, several evidence (Xiong et al., 2020; Kedia et al., 2024) suggest that Post-LN can degrade gradient flow in deeper networks, leading to vanishing gradients and slower convergence.

**Pre-LN.** The Pre-Layer Normalization (Pre-LN) (Dubey et al., 2024) scheme, normalization is applied to the module's input *before* processing:

$$y_l = x_l + \text{Module}\big(\text{Norm}(x_l)\big). \qquad (2)$$

As for Llama 3 architecture, a final LN is applied to the network output. Pre-LN improves gradient flow during backpropagation, stabilizing early training (Xiong et al., 2020). Nonetheless, in large-scale Transformers, even Pre-LN architectures are not immune to instability during training (Wortsman et al., 2024; Zhai et al., 2023). As shown in Figure 2, unlike Post-LN—which places LN at position $C$—Pre-LN, which places LN only at position $A$, can lead to a "highway" structure that is continuously maintained throughout the entire model if the module produces an output with a large magnitude. This phenomenon might be related to the "massive activations" observed in trained models (Sun et al., 2024; Fishman et al., 2024).

## 3.2. Variance Behavior from Initialization to Training

As discussed by Xiong et al. (2020) and Kedia et al. (2024), Transformer models at *initialization* exhibit near-constant hidden-state variance under Post-LN and linearly increasing variance under Pre-LN. Most of the previous studies have concentrated on this early-stage behavior. However, Recent

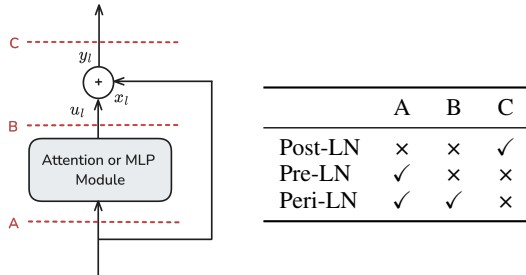

[2]Unless stated otherwise, LN refers to both LayerNorm (Ba et al., 2016) and RMSNorm (Zhang & Sennrich, 2019).

[3]We use "hidden state" and "activation" interchangeably.

| | A | B | C |
|---|---|---|---|
| Post-LN | × | × | ✓ |
| Pre-LN | ✓ | × | × |
| Peri-LN | ✓ | ✓ | × |

*Figure 2.* Placement of normalization in Transformer sub-layer.

studies have also reported large output magnitudes in both the pre-trained Attention and MLP modules (Dehghani et al., 2023; Wortsman et al., 2024; Fishman et al., 2024). To bridge the gap from initialization to the fully trained stage, we extend our empirical observations in Figure 1 beyond initial conditions by tracking how these variance trends evolve at intermediate points in training.

During training, we find that Post-LN maintains a roughly constant variance, which helps avert exploding activations. However, as models grow deeper and training progresses, consistently normalizing $x_l + \text{Module}(x_l)$ can weaken gradient flow, occasionally causing partial vanishing gradients and slower convergence. In contrast, Pre-LN normalizes $x_l$ before the module but leaves the module output unnormalized, allowing hidden-state variance to accumulate exponentially once parameter updates amplify the input. Although Pre-LN preserves gradients more effectively in earlier stages, this **exponential** growth in variance can lead to "massive activations" (Sun et al., 2024), risking numeric overflow and destabilizing large-scale training.

**Takeaways from Pre-LN & Post-LN.** (1) *Keeping the Highway Clean: Post-LN's Potential for Gradient Vanishing and Slow Convergence.* When layer normalization is placed directly on the main path (Placement $C$ in Figure 2), it can cause gradient vanishing and introduce fluctuations in the gradient scale, potentially leading to instability (Xiong et al., 2020). (2) *Maintaining a Stable Highway: Pre-LN May Not Suffice for Training Stability.* Pre-LN does not normalize the main path of the hidden states, thereby avoiding the issues that Post-LN encounters. Nevertheless, a structural characteristic of Pre-LN is that any large values arising in the Attention or MLP modules persist through the residual identity path. In particular, as shown in Figure 1, the *exponentially* growing magnitude and variance of the hidden states in the forward path may lead to numerical instability.

## 3.3. Peri-Normalization in Transformers

Recent open-source Transformer architectures have placed normalization layers in unconventional placements (Rivière et al., 2024; Team et al., 2025; OLMo et al., 2024). In particular, these models apply an additional normalization

layer at the module output (Output-LN), yet the benefits of this design choice remain unclear. To assess the impact of Output-LN, we analyze the Peri-LN architecture.

**Peri-LN.** The Peri-Layer Normalization (Peri-LN) applies LN twice within each layer—before and after the module—and further normalizes the input and final output embeddings. Formally, for the hidden state $x_l$ at layer $l$:

1. *(Optional) Initial Embedding Normalization:*

$$y_o = \text{Norm}(x_o),$$

2. *Input- & Output-Normalization per Layer:*

$$y_l = x_l + \text{Norm}\Big(\text{Module}\big(\text{Norm}(x_l)\big)\Big), \quad (3)$$

3. *Final Embedding Normalization:*

$$y_L = \text{Norm}(x_L),$$

where $x_o$ denotes the output of the embedding layer, the hidden input state. $y_0$ represents the normalized input hidden state. $x_L$ denotes the hidden state output by the final layer $L$ of the Transformer sub-layer. This design unifies pre- and output-normalization to regulate variance from both ends. For clarity, the locations of normalization layers in the Post-, Pre-, and Peri-LN architectures are illustrated in Figure 2.

Both the latest Gemma (Team et al., 2025; Rivière et al., 2024) and OLMo (OLMo et al., 2024) model families, which apply output layer normalization, adopt the same peri-normalization strategy. However, neither line of work rigorously examines how this placement constrains variance or mitigates large residual activations. Our study extends these open-sourced large-scale models by providing both theoretical and empirical insights into the Peri-LN scheme.

**Controlling Variance & Preserving Gradients.** By normalizing both the input and output of each sub-layer, Peri-LN constrains the *residual spikes* commonly observed in Pre-LN, while maintaining a stronger gradient pathway than Post-LN. Concretely, if $\text{Norm}(\text{Module}(\text{Norm}(x_l)))$ exhibits near-constant variance $\beta_0$, then

$$\text{Var}(x_{l+1}) \approx \text{Var}(x_l) + \beta_0, \quad (4)$$

resulting in *linear or sub-exponential* growth of activations, in contrast to the exponential growth patterns of Pre-LN.

Although Pre-LN and Peri-LN exhibit comparable, roughly linear variance growth at initialization (De & Smith, 2020; Xie et al., 2023), their trajectories diverge once training begins. The additional normalization layer (Output-LN)

in Peri-LN preserves the conditions of Eq. 4, enabling the model's hidden states to remain better conditioned. By contrast, the rapid surge in variance observed in Pre-LN can trigger instability during the early stages of training, an effect we quantify empirically in Sections 5 and 6.

### 3.4. Stability Analysis in Normalization Strategies

Xiong et al. (2020) showed that, *at initialization*, Pre-LN exhibits smaller gradient scales at the final layer compared to Post-LN, with respect to model depth. In this study, we broaden our analysis beyond initialization to monitor hidden state variance over the full course of training. Building on the earlier observation that the deepest layer exhibits the largest activations, we focus on this surge in the final layer under the Peri-LN strategy to clarify its impact on training stability. To this end, we analyze stability by examining the gradient norm with respect to the final layer weights in the presence of massive activation. Formal statements and detailed proofs are presented in Appendix C.

**Proposition 3.1** (Informal). *Let $\mathcal{L}(\cdot)$ be the loss function, and let $W^{(2)}$ denote the weight of the last layer of $\text{MLP}(\cdot)$. Let $\gamma$ be the scaling parameter in $\text{Norm}(\cdot)$, and let $D$ be the dimension. Then, the gradient norm for each normalization strategy behaves as follows.*

*(1) Pre-LN (exploding gradient). Consider the following sequence of operations:*

$$\tilde{x} = \text{Norm}(x), a = \text{MLP}(\tilde{x}), o = x + a, \quad (5)$$

*then*

$$\left\| \frac{\partial \mathcal{L}(o)}{\partial W_{i,j}^{(2)}} \right\| \propto \|h_i\|, \quad (6)$$

*where $h := \text{ReLU}\left(\tilde{x}W^{(1)} + b^{(1)}\right)$. In this case, when a massive activation $\|h\|$ occurs, an exploding gradient $\|\partial\mathcal{L}/\partial W^{(2)}\|$ can arise, leading to training instability.*

*(2) Peri-LN (self-regularizing gradient). Consider the following sequence of operations:*

$$\tilde{x} = \text{Norm}(x), a = \text{MLP}(\tilde{x}), \tilde{a} = \text{Norm}(a), o = x + \tilde{a}, \quad (7)$$

*then*

$$\left\| \frac{\partial \mathcal{L}(o)}{\partial W_{i,j}^{(2)}} \right\| \leq \frac{4\gamma\sqrt{D}\,\|h\|}{\|a\|}, \quad (8)$$

*where $h := \text{ReLU}\left(\tilde{x}W^{(1)} + b^{(1)}\right)$. In this case, even when a massive activation $\|h\|$ occurs, $\text{Norm}(\cdot)$ introduces a damping factor $\|a\|$, which ensures that the gradient norm $\|\partial\mathcal{L}/\partial W^{(2)}\|$ remains bounded.*

The layer-wise amplification documented in §3.2, combined with the bounds in Proposition 3.1, naturally explains the

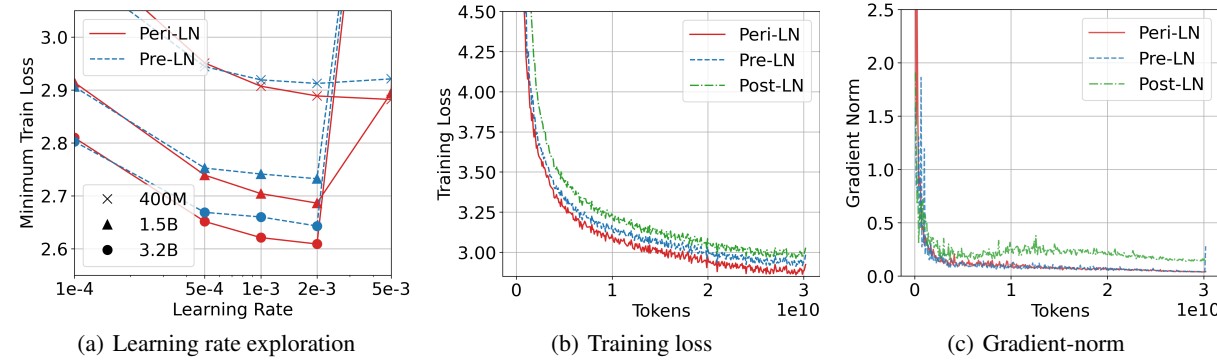

*Figure 3.* Performance comparison of Post-, Pre-, and Peri-LN Transformers during pre-training. Figure 3(a) llustrates the pre-training loss across learning rates. Pre-training loss and gradient norm of best performing 400M size Transformers are in Figure 3(b) & 3(c).

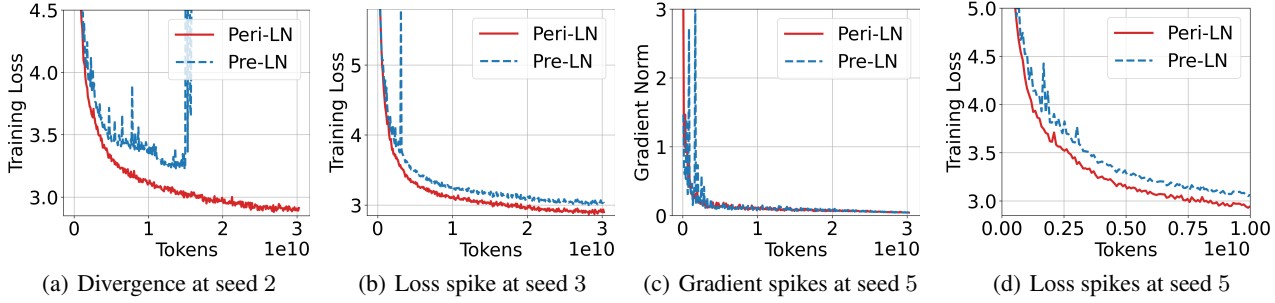

*Figure 4.* Common case of early stage instability in pre-training. In most of our experiments across different random seeds, the Pre-LN architecture exhibited early-stage instability. Although we initially suspected that a high learning rate might be the root cause, lowering it did not substantially mitigate these issues. By contrast, under the same settings, Peri-LN displayed stable training curves.

gradient spikes, and occasional divergences that arise in Pre-LN during large-scale pre-training. We revisit this phenomenon in §4.3. By contrast, the additional normalization in Peri-LN acts as a self-regularizing mechanism that damps variance growth, making the architecture less sensitive to large activations and therefore more stable in practice. The formal analysis for Post-LN is deferred to Appendix B.

# 4. Experiments

In this section, we provide a comprehensive comparison of Post-, Pre-, and Peri-Layer Normalization (LN) across large-scale Transformer pre-training, instruction-tuning, and subsequent evaluations on the language domain.

## 4.1. Experimental Setting

Excluding the embedding parameters, the model size is set to the parameters 400M, 1.5B and 3.2B, respectively. Each model is trained on 30 billion tokens. To ensure reliable validation, we pre-train each model with *five different training seeds* in all experiments. We perform a exploration of the learning rates, ranging from $1 \times 10^{-4}$ to $5 \times 10^{-3}$ to identify the U-shaped pattern for each LN strategy. The sequence length is set to 8192, and the weight decay coefficient is fixed at 0.033. We employ Megatron-LM[4] to

pre-train the Transformers under each LN strategy. We use the DCLM-baseline dataset (Li et al., 2024a), along with the "cl100k_base" version of the TikToken tokenizer[5]. Unless otherwise noted, most training and model configurations follow those of the DCLM experiments(Li et al., 2024a). For normalization layer, we primarily employ RMSNorm. Further details are in Appendix D.

## 4.2. Pre-Training Large Language Models

Figure 3(a) illustrates the pre-training loss across learning rates for models ranging in size from 400M to 3.2B parameters. Notably, the Peri-LN architecture consistently achieves superior loss curves over this entire model size. Since Pre-LN shows best performance at learning rate $2 \times 10^{-3}$ across all model size, we set this to the default learning rate for Pre-LN and Peri-LN. Unlike Pre-LN, Post-LN's appropriate learning rate lies in a lower range, so we provide a separate summary in Appendix E.1. In Figures 3(b) and 3(c), we compare the pre-training loss and the gradient norm curve at each LN strategy's best-performing learning rate of 400M size models. The same trend is observed across different model sizes (§E). In particular, when we sweep over training seeds and learning rates, Pre-LN frequently exhibits spikes in the gradient-norm curve, whereas Peri-LN shows comparatively few, thereby supporting Proposition 3.1.

---

[4]https://github.com/NVIDIA/Megatron-LM

[5]https://github.com/openai/tiktoken

*Table 1.* Average benchmark scores (with standard deviations) across 5 different training seeds for Post-, Pre-, and Peri-Layer Normalization language models. Each model size excludes the embedding parameters. *Loss* denotes the evaluation loss on random samples of the C4 dataset. *Arch.* denotes architecture, and *Avg.* denotes the averaged benchmark score across tasks. *SFT avg.* denotes the averaged benchmark score across tasks of instruction fine-tuned models. Diverged checkpoints are excluded from the evaluation score computation.

| Size | Arch. | ARC-Easy | HellaSwag | PIQA | SIQA | Winogrande | Avg. ↑ | Loss ↓ | SFT Avg. ↑ |
|------|-------|----------|-----------|------|------|------------|--------|--------|------------|
| 400M | Post-LN | 35.70 ±1.09 | 28.91 ±0.16 | 62.26 ±0.73 | 34.48 ±1.04 | 50.88 ±0.75 | 42.45 | 7.46 | 46.44 |
|      | Pre-LN | 54.87 ±1.63 | 34.17 ±1.66 | 68.79 ±1.34 | 39.73 ±0.59 | 50.88 ±2.35 | 49.69 | 3.43 | 49.96 |
|      | Peri-LN | **57.51** ±0.81 | **37.46** ±0.34 | **69.48** ±0.39 | **40.64** ±0.51 | **52.74** ±0.67 | **51.57** | **3.34** | **51.96** |
| 1.5B | Post-LN | 42.92 ±0.93 | 31.69 ±0.41 | 66.72 ±0.40 | 35.84 ±0.61 | 50.30 ±1.87 | 45.49 | 5.38 | 48.95 |
|      | Pre-LN | 61.51 ±1.22 | 39.88 ±1.53 | 71.41 ±0.88 | 41.23 ±0.97 | 54.51 ±2.07 | 53.71 | 3.29 | 53.89 |
|      | Peri-LN | **66.17** ±0.21 | **43.94** ±0.34 | **73.63** ±0.24 | **42.34** ±0.83 | **56.64** ±0.44 | **56.55** | **3.18** | **56.94** |
| 3.2B | Post-LN | 45.30 ±3.23 | 33.59 ±0.44 | 66.45 ±2.86 | 35.82 ±1.09 | 51.10 ±1.60 | 46.45 | 4.43 | 49.33 |
|      | Pre-LN | 65.24 ±2.32 | 44.23 ±2.32 | 73.86 ±1.19 | 42.68 ±0.07 | 57.42 ±2.51 | 56.69 | 3.20 | 57.08 |
|      | Peri-LN | **68.73** ±0.57 | **46.99** ±0.21 | **74.31** ±0.41 | **43.00** ±0.73 | **59.76** ±0.78 | **58.56** | **3.11** | **59.02** |

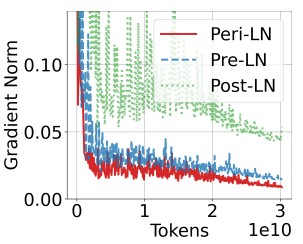
(a) Gradient-norm at seed 5

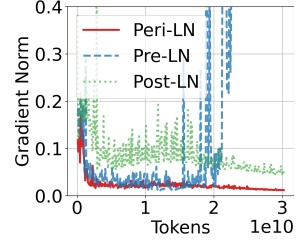
(b) Gradient-norm at seed 4

*Figure 5.* Final-layer gradient norms for seeds 4 and 5.

### 4.3. Early Stage Instability in Pre-Training

Early in pre-training, Pre-LN models consistently show gradient spikes, loss surges, and occasional divergence across seeds and scales (Fig. 4). These issues are far less pronounced in Peri-LN. We posit that the instability of Pre-LN arises from three factors: (1) the hidden state variance exhibits a sudden surge from initialization through the early stages of optimization, deviating from the linear trend predicted by Eq. 4 (see §3.3); (2) the exponential growth of hidden state variance across both depth and training steps; and (3) the instability caused by the massive activations (Proposition 3.1). Among these, we highlight the variance growth along the main path as the principal driver of the observed divergence. To corroborate this, Section 6 presents targeted experiments that manipulate weight decay and weight initialization schemes, demonstrating how curbing extreme variance mitigates the instability of each LN strategy. The curves in 4(a), 4(b), and 4(c) are from a 400M model, whereas 4(d) corresponds to a 1.5B model.

### 4.4. Gradient Norm of the Final-layer

Motivated by Proposition 3.1, we track the final-layer gradient-norm in two representative runs selected from five training seeds. Figure 5(a), now including the newly added Peri-LN results, confirms the hierarchy reported by Xiong et al. (2020): whenever training remains stable, the gradient norms satisfy Post-LN > Pre-LN > Peri-LN. However, Figure 5(b) shows a run in which the Pre-LN model diverges even though every hyperparameter matches the stable run

except for the random seed. In Section 5.1, we examine this failure in greater depth and relate it to Proposition 3.1. The curves in Figure 5 are obtained from 400M models.

### 4.5. Benchmark Evaluations & Instruction Tuning

To evaluate how well the pre-training loss aligns with its benchmark performance, we conduct five separate benchmark evaluations. Furthermore, to investigate instruction-following capabilities under different layer normalization strategies, we conduct additional training using the LIMA dataset (Ouyang et al., 2022; Zhou et al., 2023). Diverged checkpoints are excluded from the evaluation score computation (mostly occurs in Pre-LN). Additional training hyperparameters for SFT are given in Appendix D.2. As shown in Table 1, Peri-LN consistently demonstrates superior performance across all model sizes. Additionally, we note that, beyond the improved scores, the standard deviation of the benchmark results across different training seeds is reduced by more than half with Peri-LN. From this, we observe that Peri-LN *helps maintain consistency* not only in gradient stability and final loss but also in benchmark performance. For the evaluation loss, we used 10K random samples from the C4 dataset (Raffel et al., 2020). Detailed settings and individual benchmark scores are provided in Appendix J.

## 5. Analysis

Despite emerging evidence that Peri-LN outperforms Post- and Pre-LN, key uncertainties remain: *How* do different LN placements shape hidden-state statistics and gradient flow (§5.1, §5.2)? *What* role does the Output-LN scale parameter $\gamma$ play (§5.3)? And *why* does Peri-LN produce more distinctive representations than its counterparts (§5.4)? The subsections below tackle these questions in turn.

### 5.1. Growth of Hidden State

To examine in greater depth how Peri-LN affects forward propagation, we analyze the absolute magnitude and variance of the hidden states using $1,000$ samples from the

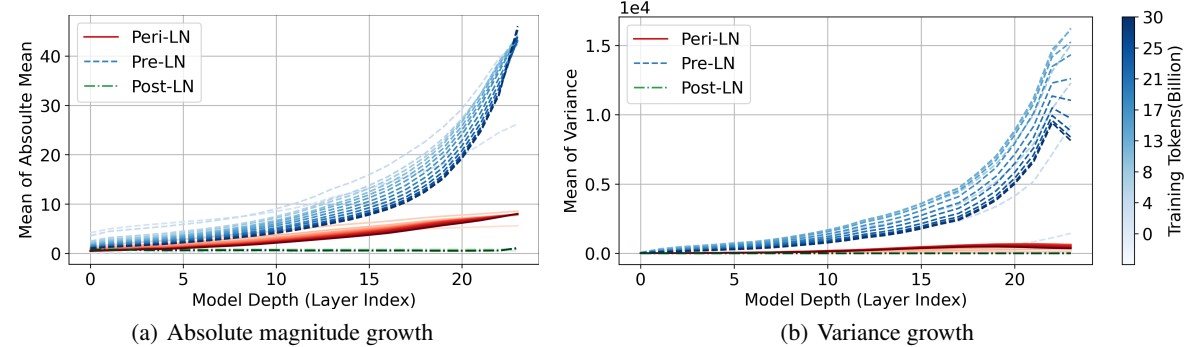

(a) Absolute magnitude growth                 (b) Variance growth

*Figure 6.* Forward hidden state growth patterns for each LN strategy in a 1.5B-parameter Transformer.

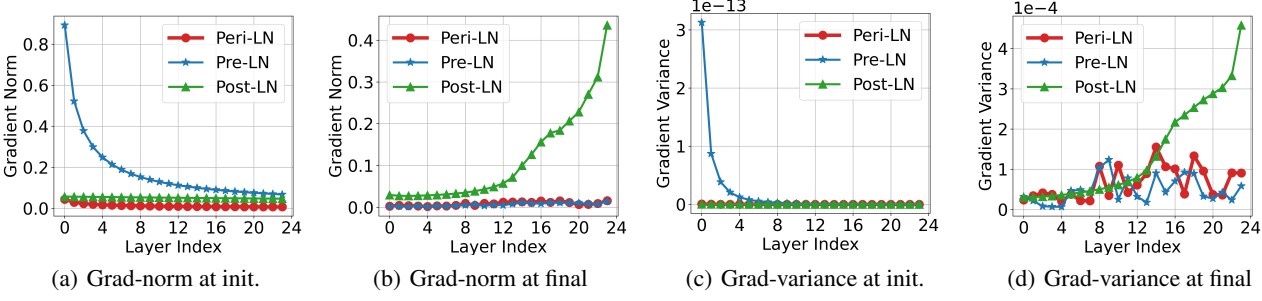

(a) Grad-norm at init.       (b) Grad-norm at final       (c) Grad-variance at init.       (d) Grad-variance at final

*Figure 7.* Backward gradient norm and variance of 1.5B Post-, Pre-, and Peri-LN Transformers at initialization (*init.*) and final training.

Wikitext dataset (Merity et al., 2016). Figure 6 shows how different normalization strategies influence forward-path hidden states over the course of training and across model depth. We observe the same pattern across all models trained with five different random seeds (§F).

Across layers, Post-LN maintains stable hidden state magnitudes and variances because the main path includes a normalization layer. In contrast, Pre-LN omits normalization after each attention and MLP sub-layer, so the magnitude and variance of the hidden states grow exponentially after the residual addition. For Peri-LN, which adds an Output-LN, these statistics remain comparatively well controlled. Across training iterations, Post-LN's block-level normalization continues to suppress large shifts, preventing substantial drift in magnitude or variance. Pre-LN starts with an approximately linear variance profile at initialization but escalates exponentially to extremely large values as optimization proceeds. Peri-LN again exhibits only moderate fluctuations, owing to Output-LN's consistent regulation of hidden-state statistics. Further discussion appears in Section 6.

### 5.2. Layer-wise Gradient Norm & Variance

Ensuring a uniform gradient flow in large-scale model training is crucial for balanced learning across the entire network (Yang & Hu, 2021; Yang et al., 2024). As shown in Figure 7, in Post-LN, gradients decrease as they propagate backward through the layers in the final stage of training, which can lead to vanishing gradients in lower-index layers. In Pre-LN, gradients increase as they propagate backward through the

layers at initialization, potentially causing explosive gradients in the early phase of training. Both strategies display non-uniform gradient distributions—either vanishing or exploding—at different stages of training. On the other hand, Peri-LN demonstrates a consistent, layer-wise gradient distribution at both initialization and the end of training. By maintaining comparatively uniform gradients with lower variance across layers, Peri-LN avoids the extremes of vanishing or exploding behaviors. This stability is particularly beneficial in deeper architectures, where balanced gradient flow is essential for effective backpropagation.

### 5.3. Learnable Parameter $\gamma$ of RMSNorm

To investigate the impact of module output normalization on training stability, as proposed in the Proposition 3.1, we fix the learnable parameter $\gamma$ of RMSNorm to 1, isolating the effect of normalization. As illustrated in Figure 8, adding output normalization to each sub-layer suppresses gradient spikes and lowers the loss relative to Transformers that employ only pre-normalization. Nonetheless, we also confirm that allowing $\gamma$ to be learnable yields slightly better performance. The trend persists consistently across model scales and random seeds. In this experiment, we omit Peri-LN's embedding layer normalization in order to isolate and evaluate the precise role and benefits of output-LN.

### 5.4. Hidden State Representation

To assess hidden state redundancy after training, we employ angular distance (Li et al., 2024b), which quantifies how sim-

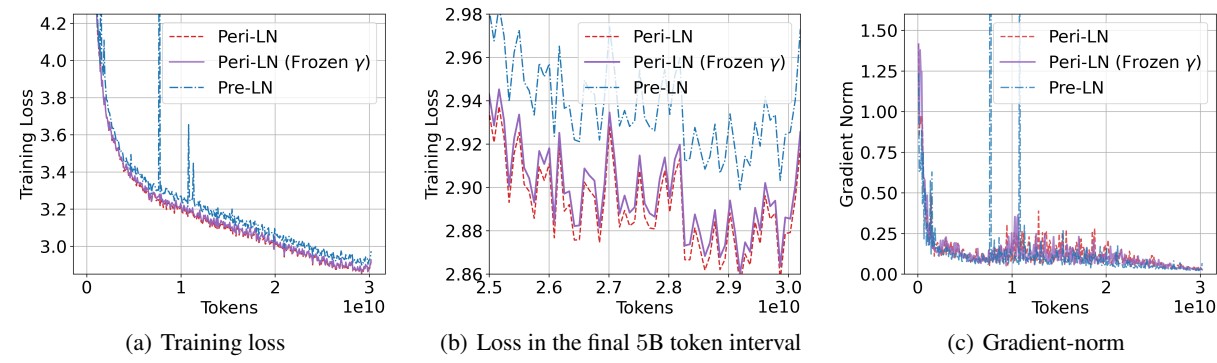

(a) Training loss      (b) Loss in the final 5B token interval      (c) Gradient-norm

*Figure 8.* Freezing learnable parameter $\gamma$ of output normalization layer in Peri-LN. we set $\gamma$ to its initial value of 1 and keep it fixed.

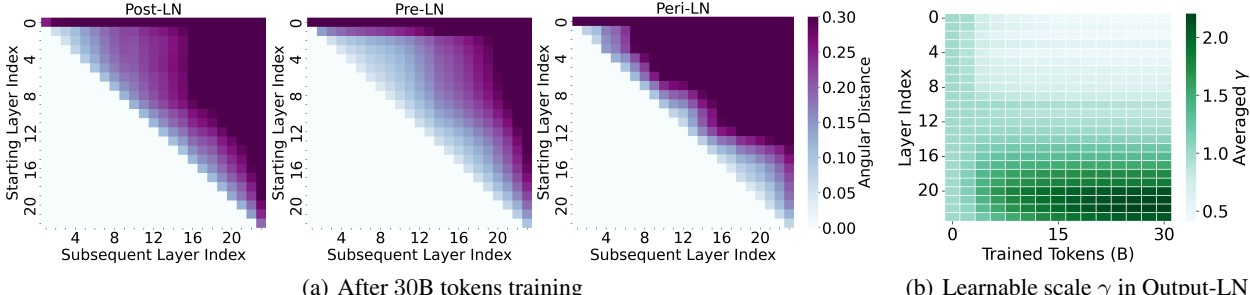

(a) After 30B tokens training      (b) Learnable scale $\gamma$ in Output-LN

*Figure 9.* Angular distance between hidden states after training. Fig. 9(b) monitor $\gamma$ of every Output-LN in Peri-LN during training.

ilar or distinct the layer representations are. As Figure 9(a) illustrates, Pre-LN produces markedly more redundant hidden states than the other variants by the end of training. We attribute this effect to the exponential growth of the main residual path in Pre-LN, which diminishes the relative contribution of individual sub-layers. In contrast, Peri-LN retains an identity path whose learnable scale begins near 1 and gradually adjusts with depth (Figure 9(b)), thereby moderating redundancy. These observations highlight the role of module-output normalization in controlling hidden state similarity. All statistics are computed on 256 random samples from RedPajama-Data-1T (Computer, 2023). Appendix L includes additional figures and initialization comparisons.

## 6. Ablation Study

To probe massive activations across conditions, we sweep weight decay coefficient and initialization variance for both Pre- and Peri-LN models, holding other settings fixed. Per-run results and detailed settings are in Appendix G.8.

### 6.1. Weight Decay

In Figure 10(a), stronger L2 regularization markedly lowers the variance curve, confirming that heavier weight decay directly curbs forward-path explosions in Pre-LN. In contrast, the same increase in weight decay reduces Peri-LN's variance growth only marginally. We take the stable run initialized with seed 3 (Table 11) and sweep the weight decay coefficient. Table 7 in the Appendix further shows that,

irrespective of the presence of massive activations, Peri-LN achieves better performance than Pre-LN.

To further probe stability under varying degrees of massive activation, we replicate the previously divergent run (seed 4) and repeat the same weight decay sweep. As Figure 10(b) demonstrates, raising the weight decay coefficient from the baseline 0.033 to 0.33 (a tenfold rise) prevents divergence, providing empirical support for Proposition 3.1. Nevertheless, strong weight decay can stabilize Pre-LN, it still fails to close the performance gap relative to Peri-LN.

### 6.2. Weight Initialization

As the initialization variance increases, the severity of massive activations rises correspondingly for Pre-LN (Figure 10(c)); at the largest variance, the model diverges outright (Appendix G.8.3). Pre-LN therefore displays marked sensitivity to its initial conditions. In contrast, across the same range of ablations, Peri-LN's loss curves and activation variances shift only marginally. This hyperparameter-insensitive robustness is corroborated by the low downstream standard deviations reported in Table 1.

### 6.3. Additional Results

For brevity, we defer an extensive set of supplementary experiments to the appendix. Appendix G reports the core robustness checks: replacing RMSNorm with LayerNorm, varying sequence lengths, reducing pre-training budgets, and ablating embedding-level normalization. OLMo2-style

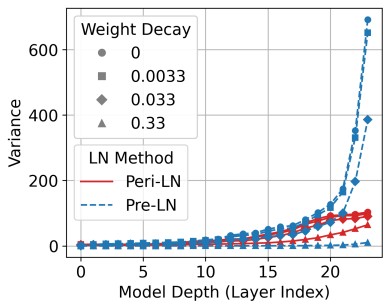 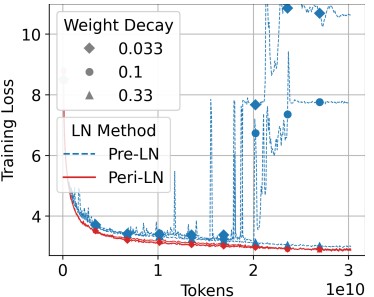 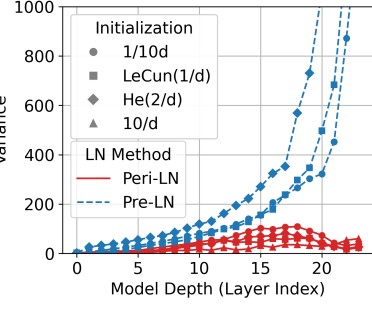

(a) Variance growth by weight decay     (b) Training loss by weight decay     (c) Variance growth by weight init.

*Figure 10.* Effects of weight decay and initialization (init.) on massive activations. 10(a) Strong weight decay relieves the variance explosion in Pre-LN. 10(b) Strong weight decay (0.33, which is $10\times$ the baseline.) suppresses Pre-LN divergence. 10(c) Smaller-scale initialization slightly curbs Pre-LN variance, while Peri-LN remains bounded regardless. $d$ denotes the model's hidden dimensionality.

Peri-LN pre-training runs appear in Appendix H. Stochastic gradient descent (SGD) baselines are summarized in Appendix I. Alternative LN placements, extending Figure 2, are provided in Appendix G.6. Across all settings, the results are consistent with the trends presented in the main Section 4 and 5, further substantiating our conclusions.

## 7. Implications

This section integrates our findings into practical guidance on variance-driven stability and precision constraints in large-scale Transformers.

### 7.1. Mitigating Variance-Driven Instability via Peri-LN

Pre-LN, the prevailing normalization strategy, is inherently prone to unchecked growth in activation variance (§5), which in turn induces numerical instability during training. Our extensive empirical analysis shows that Peri-LN—which normalizes the outputs of the Attention and MLP sub-layers—markedly curbs this variance and often prevents divergence (§4 & §6). Proposition 3.1 formalizes how excessive variance amplifies gradient norms, clarifying its causal role in destabilizing large-scale pre-training. In Pre-LN, instability is further exacerbated when the statistical conditions assumed at initialization depart markedly from those observed later in training (§3.3). By contrast, Peri-LN alleviates this discrepancy, thereby improving training stability, and delivering additional performance gains.

### 7.2. Precision Constraints Imposed by Pre-LN

Both Pre-LN and Peri-LN architectures leave the main hidden state path unnormalized, so once large values arise in earlier layers, they persist through to later layers. Consequently, Pre-LN's additive residual path might generates activations near or beyond the FP16 limit. To gauge how often these values exceed FP16 yet remain within BF16, we track the top-100 absolute hidden state values for 3.2B-parameter Pre-LN and Peri-LN models. In Figure 11, the blue band

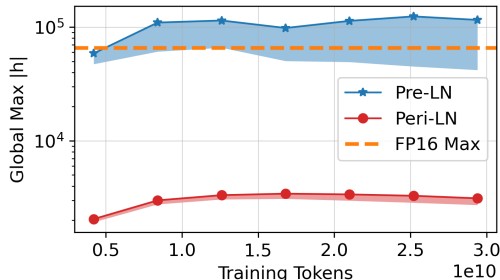

*Figure 11.* Evolution of extreme hidden state absolute magnitudes during training. Colored bands trace the range of the global top-100 absolute activations. Pre-LN (blue) quickly surpasses the FP16 representable maximum, while Peri-LN (red) stays below it.

for Pre-LN surpasses the FP16 maximum bound as early as 0.5B training tokens whereas Peri-LN (red band) consistently remains below this threshold. This pattern, echoing Sun et al. (2024), highlights that choosing FP16 or BF16 is not just a hardware preference but is closely linked to how hidden state magnitudes evolve within the model. Earlier work on OPT (Zhang et al., 2022), which was pre-trained using FP16 precision, suggests that the training instabilities they observed were likely exacerbated by numerical overflows and gradient pathologies (Proposition 3.1) arising when activations exceeded the representable range of FP16.

## 8. Conclusion

We explore the placement of layer normalization within the Transformer architecture to better understand its role during training. By systematically comparing Post-LN, Pre-LN, and newly termed Peri-LN, we highlight their distinct impacts on stability, final performance, and optimization dynamics. Our findings suggest that placing LN on module outputs in addition to the Pre-LN can help manage large activations while preserving beneficial gradient flow, thereby offering a promising balance for stable optimization. By unifying these approaches under the term *Peri-LN*, we seek to consolidate existing variants and encourage deeper investigation into this underexplored alternative.

## Acknowledgements

We thank our colleague Jeongin Bae for inspiring the underlying motivation for this research. We are also grateful to Jung Hyun Lee, Seonghyeon Kim, and Seunghyun Seo for their valuable assistance during the early stages discussions. Finally, we extend our gratitude to Gichang Lee, Lead of the Backbone Mission at NAVER Cloud, for his unwavering support. This work was partly supported by Institute for Information & communications Technology Technology Planning & Evaluation(IITP) grant funded by the Korea government(MSIT)(RS-2019-II190075, Artificial Intelligence Graduate School Support Program(KAIST), No.RS-2021-II212068, Artificial Intelligence Innovation Hub, No. RS-2024-00509279, Global AI Frontier Lab)

## Impact Statement

The rapid advancement of Transformer-based large language models (LLMs) has enabled remarkable breakthroughs in natural language understanding and generation. However, these models also pose significant challenges, including concerns around safety, bias, and the computational cost associated with large-scale training. As LLMs become increasingly integral to various AI applications, ensuring their stability, efficiency, and accessibility remains a critical research focus.

Our work addresses these challenges by proposing a more stable and cost-effective large-scale training methodology. By improving training efficiency and reducing the associated computational overhead, we lower the barrier to entry for organizations seeking to develop or fine-tune foundation models. This democratization of LLM technology fosters broader participation in AI research and development, accelerating innovation while mitigating concerns over resource concentration in a few major players. Given the growing industry focus on optimizing LLM deployment costs, our contributions are particularly relevant in the current AI research landscape.

Improving the cost-effectiveness of large-scale training simultaneously lowers AI's environmental footprint by reducing the vast energy consumption and carbon emissions inherent in state-of-the-art LLM development. This efficiency not only aligns with global sustainability goals but also enables smaller research labs and academic groups to pursue cutting-edge AI without prohibitive resource demands, fostering a more inclusive and responsible ecosystem.

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

s.cc/paper_files/paper/2023/hash/ac6
62d74829e4407ce1d126477f4a03a-Abstrac
t-Conference.html.

# A. Related Work

**Activation Dynamics in Large Language Models.** Studies on the distribution and magnitude of activations in deep neural networks have revealed that certain outlier features can significantly affect model behavior and efficiency. Dettmers et al. (2022) examined Transformer architectures, highlighting how specific feature dimensions may exhibit unusually large values (outliers) that disrupt quantization and overall system performance. Extending this line of work, Sun et al. (2024) identified the occurrence of "massive activations"—extremely large activation values that persist across multiple layers. Unlike standard outliers, these massive activations remain relatively invariant to different inputs, effectively functioning as implicit bias terms in large language models (LLMs). Notably, such extreme values can skew the self-attention mechanism, causing the model to attend disproportionately to certain tokens. These observations demonstrate that even with standard normalization layers in place, hidden biases may linger in internal representations, underscoring the importance of deeper analyses of activation behavior in LLMs.

**Variance Control and Normalization in Convolutional Networks.** The interplay between activation variance and training stability has also been extensively explored in convolutional neural networks (CNNs). De & Smith (2020) showed that Batch Normalization (BN) stabilizes the training of residual networks by effectively downscaling activation variance in the residual branches, thereby improving gradient behavior. However, BN imposes certain constraints, such as dependence on batch size and additional computational overhead for estimating batch statistics. Consequently, several normalization-free or alternative normalization approaches have been investigated. For instance, Brock et al. (2021b) introduced "Normalizer-Free ResNets," which manage activation variance through learnable scaling parameters. This approach achieved competitive performance without relying on BN, highlighting the critical role of effective variance control in fostering stable optimization and strong generalization in CNNs.

**Layer Normalization in Transformers.** Training stability in Transformer architectures is closely tied to the choice and placement of layer normalization (LN). Xiong et al. (2020) reported that Transformers employing a Post-LN structure often suffer from gradient instabilities at initialization, requiring a careful learning-rate warm-up phase to mitigate these issues. In contrast, Pre-LN helps maintain more stable gradients during the early stages of training. However, Kedia et al. (2024) showed that while Post-LN can lead to vanishing or exploding gradients in deep Transformers, Pre-LN may induce hyperbolic gradient growth. These findings illustrate the nuanced trade-offs of normalization placement and draw parallels to earlier CNN studies, where careful management of activation variance proved essential for stable deep learning. Ding et al. (2021) introduced Sandwich-LN in the vision domain for the first time, yet they paid little attention to the structural characteristics and differences that distinguish one LN placement from another. In language domain, several major open-source language models (e.g., Olmo2 (OLMo et al., 2024), Gemma2 (Rivière et al., 2024), and Gemma3 (Team et al., 2025)) already employ a Peri-LN-like structure (see Section 3). Nevertheless, previous technical reports have not explained why this design might be advantageous compared with the more widely studied Pre-LN and Post-LN. By investigating Peri-LN in detail, we aim to highlight the structural benefits that appear to underlie its empirical success in these implementations.

**Gradient Propagation and Depth Scaling** Ensuring consistent gradient propagation across many layers is pivotal for stable training in very deep models. Yang & Hu (2021) (Tensor Programs IV) introduced the concept of Maximal Update Parametrization ($\mu$P) in the infinite-width regime to preserve feature learning, preventing gradients from collapsing into kernel-like dynamics. Building on this, Yang et al. (2024) (Tensor Programs VI) proposed Depth-$\mu$P, which scales residual branches and learning rates according to network depth. Their theoretical analysis indicates that improper depth-dependent scaling leads to vanishing or exploding gradients, ultimately diminishing the diversity of learned representations. These insights highlight the necessity for principled scaling strategies and careful initialization to maintain robust gradient flow in deep architectures.

**Summary.** Taken together, these studies underscore the importance of managing activation variance and hidden biases to achieve stable training and expressive internal representations in modern deep networks. In Transformer-based models, normalization choice and placement—such as Post-LN, Pre-LN, or other variants—play a significant role in controlling gradient dynamics and overall performance. While Post-LN and Pre-LN have received significant attention, we focus on a comparative analysis that includes *Peri-LN*, an alternative normalization placement that has thus far been underexplored but holds potential for enhancing training stability and model performance.

## B. Proposition 3.1 of Post-LN

**Proposition B.1.** *Post-LN (vanishing gradient). Consider the following sequence of operations:*

$$a = \text{MLP}(x), o = x + a, \tilde{o} = \text{Norm}(o), \tag{9}$$

*then*

$$\left\| \frac{\partial \mathcal{L}(\tilde{o})}{\partial W_{i,j}^{(2)}} \right\| \leq \frac{4\,\gamma\,\sqrt{D}\,\|h\|}{\|x + a\|}, \tag{10}$$

*where $h := \text{ReLU}\left(xW^{(1)} + b^{(1)}\right)$. Since Post-LN normalizes the hidden state after each residual addition along the main path, the activation norm $|h|$ tends to remain relatively moderate. As a result, in Post-LN, the gradient scale is less sensitive to the magnitude of activations and more significantly influenced by model depth, as previously analyzed by Xiong et al. (2020) and Kedia et al. (2024), leading to vanishing gradients as depth increases.*

## C. Proof of Theoretical Insight

To support the claim that Peri-LN enhances the stability of training in such cases, we analyze the gradient norm in the final layer. For simplicity, we use RMSNorm and ReLU here. Here, we assume that $\gamma$, the scaling parameter used in RMSNorm, is positive, and we empirically verified that it remains strictly positive across models of all sizes during training.

**Proposition C.1.** *Consider the following sequence of operations:*

$$\tilde{x} = \text{RMSNorm}(x),$$
$$a = \text{ReLU}(\tilde{x}W^{(1)} + b^{(1)})W^{(2)} + b^{(2)},$$
$$o = x + a.$$

*Then,*

$$\frac{\partial \mathcal{L}(o)}{\partial W_{i,j}^{(2)}} = h_i(\hat{p}_j - y_j), \tag{11}$$

*where $h := \text{ReLU}\left(xW^{(1)} + b^{(1)}\right)$, $\hat{p} := \text{softmax}(o)$, and $y$ is the label (one-hot vector).*

*Proof.* By the chain rule,

$$\frac{\partial \mathcal{L}(o)}{\partial W_{i,j}^{(2)}} = \underbrace{\frac{\partial \mathcal{L}(o)}{\partial o}}_{\text{(a):}1\times D} \times \underbrace{\frac{\partial o}{\partial a}}_{\text{(b):}D\times D} \times \underbrace{\frac{\partial a}{\partial W_{i,j}^{(2)}}}_{\text{(c):}D\times 1}. \tag{12}$$

(a) It is known that

$$\frac{\partial \mathcal{L}(o)}{\partial o_k} = \hat{p}_k - y_k. \tag{13}$$

So,

$$\frac{\partial \mathcal{L}(o)}{\partial o} = \begin{bmatrix} \hat{p}_1 - y_1 & \hat{p}_2 - y_2 & \cdots & \hat{p}_D - y_D \end{bmatrix}. \tag{14}$$

(b) Since $o = x + a$,

$$\frac{\partial o}{\partial a} = I. \tag{15}$$

(c) Recall that

$$a := \text{ReLU}(\tilde{x}W^{(1)} + b^{(1)})W^{(2)} + b^{(2)}. \tag{16}$$

For convenience, let

$$h := \text{ReLU}(\tilde{x}W^{(1)} + b^{(1)}). \tag{17}$$

Then, we have

$$\frac{\partial a_k}{\partial W_{i,j}^{(2)}} = \frac{\partial}{\partial W_{i,j}^{(2)}} \left( \sum_{p=1}^{H} h_p W_{p,k}^{(2)} + b_k^{(2)} \right) = h_i \, \delta_{k,j}. \tag{18}$$

In vector representation,

$$\frac{\partial a}{\partial W_{i,j}^{(2)}} = \begin{bmatrix} 0 & \cdots & h_i & \cdots & 0 \end{bmatrix}^\top, \tag{19}$$

where the only nonzero entry is in the $j$-th component.

Thus, by putting these all together,

$$\frac{\partial \mathcal{L}(o)}{\partial W_{i,j}^{(2)}} = h_i (\hat{p}_j - y_j). \tag{20}$$

$$\square$$

**Proposition C.2.** *Consider the following sequence of operations:*

$$\tilde{x} = \mathrm{RMSNorm}(x),$$
$$a = \mathrm{ReLU}(\tilde{x} W^{(1)} + b^{(1)}) W^{(2)} + b^{(2)},$$
$$\tilde{a} = \mathrm{RMSNorm}(a),$$
$$o = x + \tilde{a}.$$

*Then,*

$$\left\| \frac{\partial \mathcal{L}(o)}{\partial W_{i,j}^{(2)}} \right\| \leq \frac{4\gamma \sqrt{D} \|h\|}{\|a\|}, \tag{21}$$

*where $\gamma$ is the scaling parameter used in $\mathrm{RMSNorm}(\cdot)$, $D$ is the dimensionality, and $h := \mathrm{ReLU}\left(x W^{(1)} + b^{(1)}\right)$.*

*Proof.* By the chain rule,

$$\frac{\partial \mathcal{L}(o)}{\partial W_{i,j}^{(2)}} = \underbrace{\frac{\partial \mathcal{L}(o)}{\partial o}}_{(a):1 \times D} \times \underbrace{\frac{\partial o}{\partial \tilde{a}}}_{(b):D \times D} \times \underbrace{\frac{\partial \tilde{a}}{\partial a}}_{(c):D \times D} \times \underbrace{\frac{\partial a}{\partial W_{i,j}^{(2)}}}_{(d):D \times 1}. \tag{22}$$

(a) We have

$$\left\| \frac{\partial \mathcal{L}(o)}{\partial o} \right\| = \|\hat{p} - y\| \leq \|\hat{p}\| + \|y\| = 2. \tag{23}$$

(b) We also have

$$\left\| \frac{\partial o}{\partial \tilde{a}} \right\| = \|I\| = 1. \tag{24}$$

(c) Recall that

$$\tilde{a} := \mathrm{RMSNorm}(a) = \gamma \cdot \frac{a}{\sqrt{\frac{1}{D} \sum_{k=1}^{D} a_k^2 + \epsilon}}. \tag{25}$$

Then, $\frac{\partial \tilde{a}}{\partial a}$ is the Jacobian matrix $J$ of $\mathrm{RMSNorm}(\cdot)$. For brevity, let

$$\alpha := \frac{1}{D} \sum_{k=1}^{D} (a_k)^2 . v \tag{26}$$

Then,

$$J_{p,q} = \frac{\partial \tilde{a}_p}{\partial a_q} = \gamma \cdot \frac{\partial}{\partial a_q} \left( \frac{a_p}{\sqrt{\alpha + \epsilon}} \right) \tag{27}$$

$$= \gamma \cdot \frac{1}{\sqrt{\alpha + \epsilon}} \frac{\partial a_p}{\partial a_q} + \gamma \cdot a_p \frac{\partial}{\partial a_q} \left( \frac{1}{\sqrt{\alpha + \epsilon}} \right) \tag{28}$$

$$= \gamma \cdot \frac{1}{\sqrt{\alpha + \epsilon}} \delta_{p,q} - \gamma \cdot \frac{a_p a_q}{D(\alpha + \epsilon)^{3/2}}. \tag{29}$$

In matrix representation,

$$J = \underbrace{\frac{\gamma}{\sqrt{\alpha + \varepsilon}} I}_{A} - \underbrace{\frac{\gamma}{D(\alpha + \varepsilon)^{3/2}} (a)^{\top} (a)}_{B}. \tag{30}$$

Then, we have

$$\|A\| = \left\| \frac{\gamma}{\sqrt{\alpha + \varepsilon}} I \right\| = \frac{\gamma}{\sqrt{\alpha + \varepsilon}} \|I\| = \frac{\gamma}{\sqrt{\alpha + \varepsilon}}, \tag{31}$$

and

$$\|B\| = \left\| \frac{\gamma}{D(\alpha + \varepsilon)^{3/2}} (a)^{\top} (a) \right\| = \frac{\gamma}{D(\alpha + \varepsilon)^{3/2}} \times D\alpha = \frac{\gamma \alpha}{(\alpha + \varepsilon)^{3/2}}. \tag{32}$$

So, we have

$$\|J\| = \|A - B\| \leq \|A\| + \|B\| \leq \frac{2\gamma}{\sqrt{\alpha}} = \frac{2\gamma\sqrt{D}}{\|a\|}. \tag{33}$$

(d) Since

$$\frac{\partial a}{\partial W_{i,j}^{(2)}} = \begin{bmatrix} 0 & \cdots & h_i & \cdots & 0 \end{bmatrix}^{\top}, \tag{34}$$

we have

$$\left\| \frac{\partial a}{\partial W_{i,j}^{(2)}} \right\| \leq \|h\|. \tag{35}$$

Thus,

$$\left\| \frac{\partial \mathcal{L}(o)}{\partial W_{i,j}^{(2)}} \right\| \leq 2 \times 1 \times \frac{2\gamma\sqrt{D}}{\|a\|} \times \|h\| = \frac{4\gamma\sqrt{D}\|h\|}{\|a\|}. \tag{36}$$

$\square$

**Proposition C.3.** *Consider the following sequence of operations:*

$$a = \mathrm{ReLU}(xW^{(1)} + b^{(1)})W^{(2)} + b^{(2)},$$
$$o = x + a,$$
$$\tilde{o} = \mathrm{RMSNorm}(o).$$

*Then,*

$$\left\| \frac{\partial \mathcal{L}(\tilde{o})}{\partial W_{i,j}^{(2)}} \right\| \leq \frac{4\gamma\sqrt{D}\|h\|}{\|x + a\|}, \tag{37}$$

*where $\gamma$ is the scaling parameter used in* $\mathrm{RMSNorm}(\cdot)$*, $D$ is the dimensionality, and $h := \mathrm{ReLU}\left(xW^{(1)} + b^{(1)}\right)$.*

*Proof.* The proof is analogous to the proof of the previous proposition. $\square$

# D. Detailed Experimental Setting

In this section, we provide detailed configurations of both the pretraining and supervised fine-tuning to reproduce our results.

## D.1. Configurations on Pre-Training

The common training settings are provided in Table 2. Embedding settings for the language models are listed in Table 3. For the model architecture, we primarily follow the Llama 3 architecture (Dubey et al., 2024). In the MLP module, we use SwiGLU activations. Additional details regarding the model configurations are shown in Table 4. Note that embedding parameters are excluded from the model size. Unless otherwise noted, most training and model settings follow those of the DCLM experiments (Li et al., 2024a). During the pretraining stage, each model was trained under a controlled random seed.

*Table 2.* Common configurations. *LR Schedule* denotes learning rate schedule.

| Global Batch Size | Weight Decay | Iterations | Optimizer | LR Schedule | Warmup | Weight Initialization |
|---|---|---|---|---|---|---|
| 256 | 0.033 | 14400 | Adam | Cosine | 10% | 0.02 |

*Table 3.* Embedding configurations.

| Max Position Embeddings | Position Embedding Type | Untie-embeddings-and-output-weights |
|---|---|---|
| 8192 | Rope | $True$ |

*Table 4.* Model configurations.

| Size | $n_{layers}$ | $n_{heads}$ | $d_{model}$ | $d_{head}$ |
|---|---|---|---|---|
| 400M | 24 | 8 | 1024 | 128 |
| 1.5B | 24 | 16 | 2048 | 128 |
| 3.2B | 32 | 16 | 2560 | 160 |

## D.2. Configurations on Supervised Fine-Tuning

To examine downstream task performance after instruction tuning, we employed a high-quality LIMA alignment training set consisting of $1,000$ samples (Zhou et al., 2023). Our supervised fine-tuning configuration was slightly modified from the original setup of LIMA: we fine-tuned the model for 15 epochs with a batch size of 128. The optimizer was Adam with an initial learning rate of 1e-5 and a cosine learning rate schedule. We selected the best checkpoints for each model by evaluating on OpenBookQA (Mihaylov et al., 2018), CommonSenseQA (Talmor et al., 2019), and the NLI dataset in GLUE (Wang et al., 2018).

# E. Additional Results on Pre-Training Study

### E.1. Post-Layer Normalization Architecture & Learning Rate Exploration

In order to identify the optimal performance configuration for Post-LN within the experimental setup, we conducted a learning rate exploration as shown in Figure 12. Because the appropriate learning rate for Post-LN fell into a much lower range than those for Pre-LN and Peri-LN, we treated it separately. For each Post-LN setting, the best learning rate was determined as the one yielding the lowest final training loss, with the random seed held constant during this selection process.

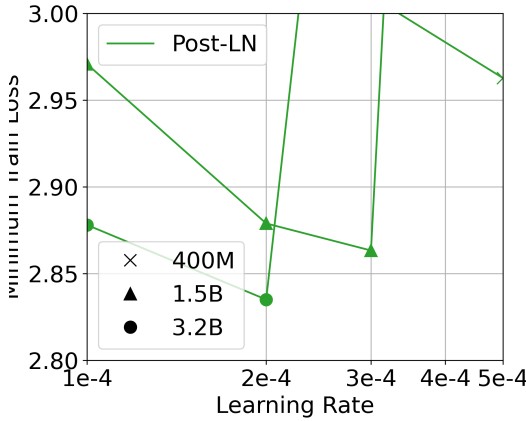

*Figure 12.* Learning rate explorations for Post-LN architectures.

### E.2. Best Performing Checkpoints Comparisons of Other Model Sizes

As an extension of Section 4.2, we present below the results for additional model sizes that were omitted previously due to space constraints. In Figures 13, we compare the pre-training loss and the gradient norm curve at each LN strategy's best-performing learning rate of 3.2B and 1.5B size models.

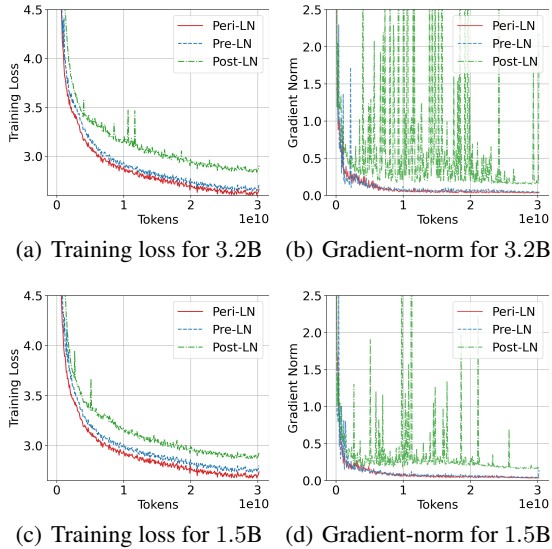

(a) Training loss for 3.2B  (b) Gradient-norm for 3.2B

(c) Training loss for 1.5B  (d) Gradient-norm for 1.5B

*Figure 13.* Performance comparison of Post-LN, Pre-LN, and Peri-LN Transformers during pre-training for other two.

# F. Additional Results on Growth of Hidden State

In this section, we examine the 400M- and 3.2B-parameter models, which were omitted in Section 5.1 due to space constraints. As illustrated in Figures 14 and 15, these models exhibit the same overall trend.

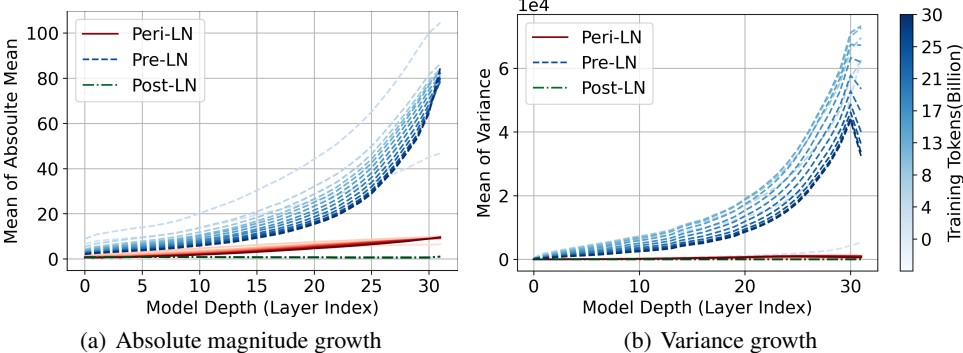

(a) Absolute magnitude growth       (b) Variance growth

*Figure 14.* The forward growth patterns of hidden state for different architectures highlight the structural impact of normalization placement. 3.2B size model.

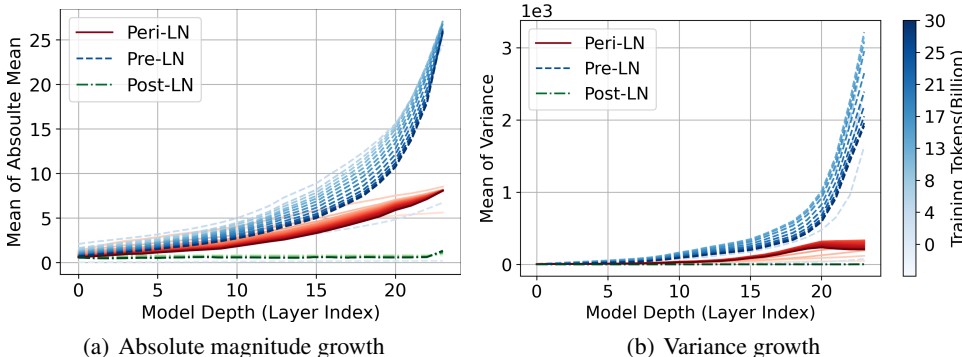

(a) Absolute magnitude growth       (b) Variance growth

*Figure 15.* The forward growth patterns of hidden state for different architectures highlight the structural impact of normalization placement. 400M size model.

# G. Additional Experimental Results on Ablation Study

## G.1. Amount of Training Tokens

In order to investigate whether the learning behavior of each LN strategy varies with the number of training tokens, we conducted an additional round of learning-rate exploration for both the Pre-LN and Peri-LN architectures. As shown in Figure 16, even as the number of training tokens increases, there is no observable shift in the optimal learning-rate range. Based on these findings, we conclude that our overall results *remain consistent*, even when the training token count is further increased. Furthermore, although a learning rate of $5 \times 10^{-3}$ leads to divergence in the smaller-scale experiments with 8B or 16B training tokens, it does not do so in the 30B-token setting. We attribute this discrepancy to the 10% warmup rate, suggesting that the warmup phase may be insufficient for the smaller-scale experiments.

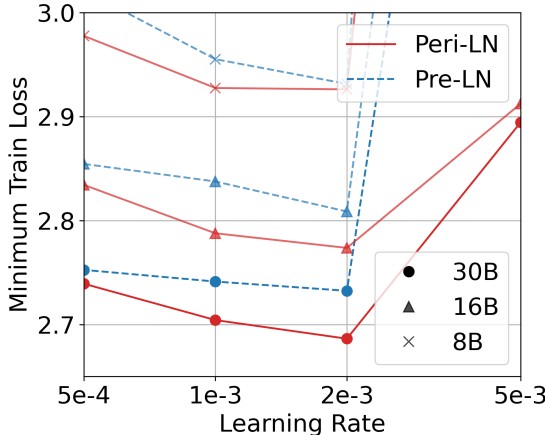

*Figure 16.* Learning rate explorations of Pre-& Peri-LN architecture with sequence length 2048 configuration.

## G.2. Sequence Length

In language models, the number of iterations per token is influenced by the sequence length, which in turn, along with the batch size, affects training statistics. We conducted an experiment to determine whether the performance trend changes when the sequence length is reduced from 8192, as set in the main text, to 2048. As shown in Figure 17, Peri-LN still surpasses Pre-LN in the learning rate exploration.

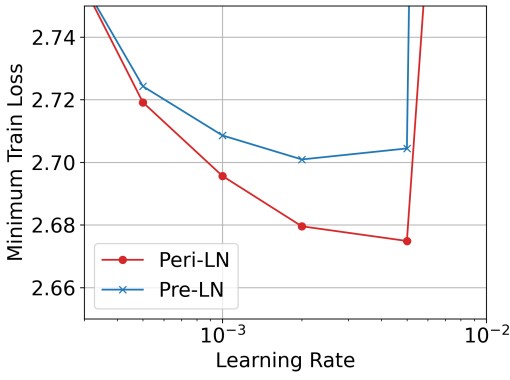

*Figure 17.* Learning rate explorations of Pre-& Peri-LN architecture with sequence length 2048 configuration.

## G.3. Warm-up

Warmup is widely recognized to influence training stability. To investigate whether a 10% warmup rate might unfairly disadvantage Pre-LN, we conducted an additional learning-rate exploration using a 30% warmup rate. As illustrated in Figure 18, the overall trend remained unchanged, and Peri-LN continued to exhibit better performance than Pre-LN in terms of loss. Furthermore, we observed that increasing the warmup rate from 10% to 30% did not reduce the frequency of gradient norm spikes in Pre-LN.

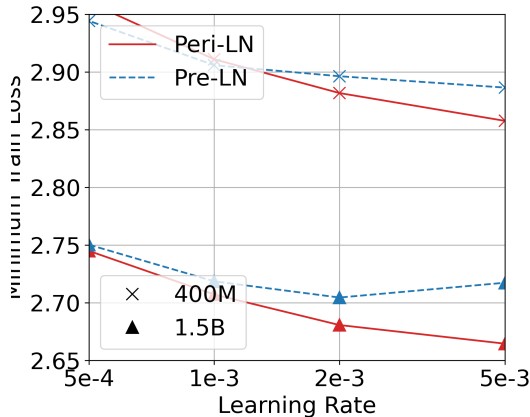

*Figure 18.* Learning rate explorations of Pre-& Peri-LN architecture with warmup 30% configuration.

## G.4. RMSNorm & LayerNorm

As illustrated in Figure 19, we conducted experiments in which RMSNorm and LayerNorm were interchanged. Consistent with the findings reported in (OLMo et al., 2024), we did not observe any notable performance differences in our RMSNorm and LayerNorm replacement experiments. Learning rate was set to 2e-3 (best performance learning rate).

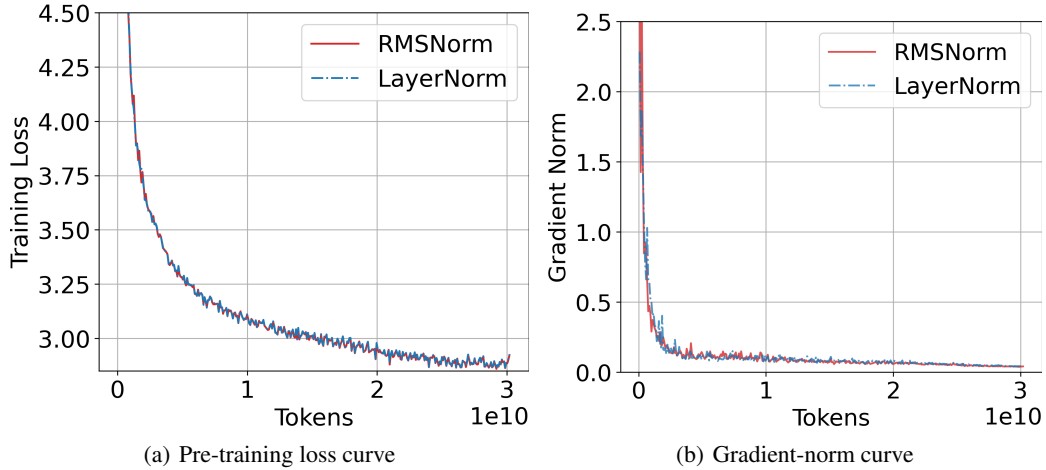

(a) Pre-training loss curve         (b) Gradient-norm curve

*Figure 19.* LayerNorm vs. RMSNorm on Peri-LN architecture. 400M size model.

### G.5. Embedding Layer Normalization of Peri-Layer Normalization Transformers

Motivated by Takase et al. (2023), we empirically explore the addition of embedding layer normalization to improve training stability and overall model performance in Transformer architectures. As illustrated in Figures 20, 21, and 22, incorporating Embedding LN in the Peri-LN architecture yields a slight improvement in pre-training loss. Furthermore, our empirical observations suggest that this effect becomes more pronounced in smaller models.

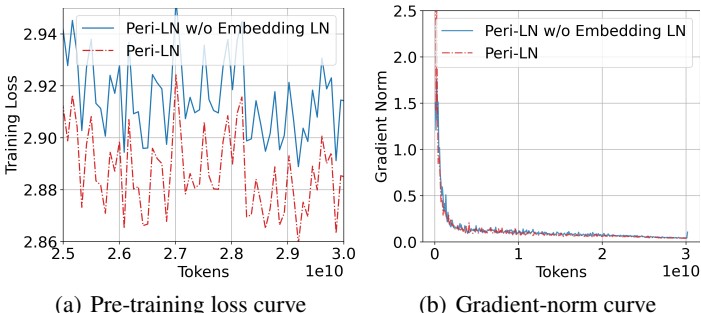

(a) Pre-training loss curve      (b) Gradient-norm curve

*Figure 20.* Loss and Gradient-norm curves comparing the presence and absence of Embedding LN in the Peri-LN architecture. 400M size model.

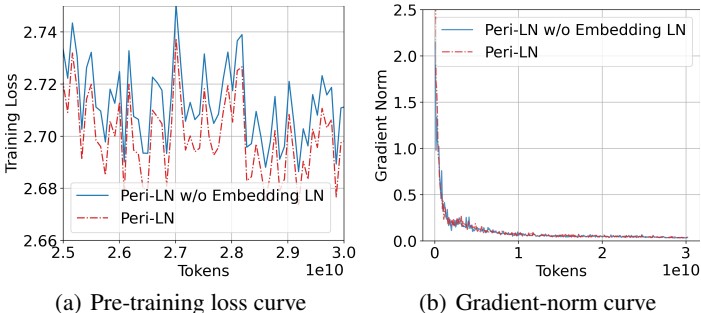

(a) Pre-training loss curve      (b) Gradient-norm curve

*Figure 21.* Loss and Gradient-norm curves comparing the presence and absence of Embedding LN in the Peri-LN architecture. 1.5B size model.

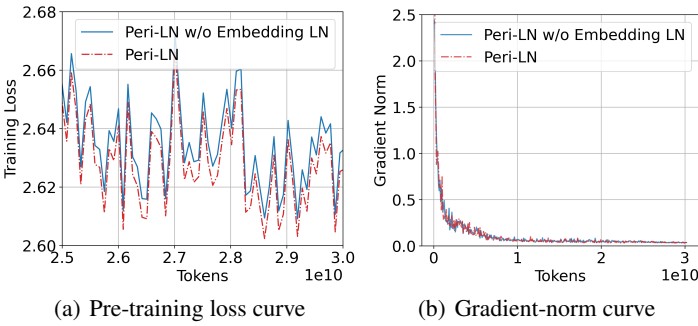

(a) Pre-training loss curve      (b) Gradient-norm curve

*Figure 22.* Loss and Gradient-norm curves comparing the presence and absence of Embedding LN in the Peri-LN architecture. 3.2B size model.

## G.6. Ablation Study on Additional Normalization Layer Placement

We additionally conduct further experiments on LN placements to compare different combinations (referred to as A, B, and C positions in Figure 2). We add configurations where LN is placed at both A + C (akin to combining Pre- and Post-LN), as well as only at B, to compare them with Peri-LN at final training loss under the controlled same training seed. We pre-train the 400M-parameter Transformers on 30B tokens each, using the same training configurations described in Section 4. As aligned with Xiong et al. (2020), our new results confirm that placing LN exclusively at C leads to training instability or suboptimal performance. In particular, the A + C configuration inherits characteristics of Post-LN (large gradient norm shifts), forcing the use of smaller learning rates and still resulting in lower overall performance than Peri-LN architecture.

*Table 5.* Final training loss and additional normalization layer placement.

| 400M | A + C | Post-LN | B | Peri-LN |
|---|---|---|---|---|
| Final Training Loss | 3.01 | 3.05 | Diverged | 2.91 |

## G.7. Peri-LN with QK-Norm

While Peri-LN alone provides robust training dynamics, QK-Norm can still enhance performance. We conducted additional experiments that confirm combining Peri-LN with QK-Norm yields slight improvements in training loss. We pre-train the 1.5B-parameter Transformers on 30B tokens each, using the same training configurations described in Section 4. As shown in Table 6, adding QK-norm to Peri-LN indeed yielded better performance, consistent with Wortsman et al. (2024). In this experiment, the Peri-LN variant equipped with QK-norm used LayerNorm instead of RMSNorm.

*Table 6.* Peri-LN with QK-Norm.

| 1.5B | Peri-LN | +QK-Norm (Wortsman et al., 2024) |
|---|---|---|
| Final Training Loss | 2.722 | 2.711 |

## G.8. Weight Decay and Weight Initialization

### G.8.1. COMMON SETTINGS

We pre-train the 400M-parameter Transformers on 30B tokens each under the controlled same training seed. We measure the training loss and averaged benchmark score for these experiments under the same evaluation settings used in Table 1 of the paper. Other configurations follow those outlined in Section 4. For the variance growth experiments in Figure 10, we adopt the same settings as in Section 5.1, except that we use 100 samples for the forward-pass statistics.

### G.8.2. WEIGHT DECAY

We conduct additional studies for various weight decay condition for both Pre-LN and Peri-LN architectures. As shown in the Table 7, Peri-LN continues to offer better performance than Pre-LN under the same settings. We provide per-run results as below:

*Table 7.* Effect of weight decay on 400M-parameter Pre-LN and Peri-LN Transformers: Final training loss and averaged benchmark score.

| 400M | Weight Decay Coefficient | 0 | 0.0033 | 0.033 | 0.33 |
|---|---|---|---|---|---|
| Final Training Loss | Pre-LN | 3.03 | 3.03 | 3.03 | 3 |
| | Peri-LN | 2.94 | 2.94 | 2.93 | 2.90 |
| Averaged Benchmark Score | Pre-LN | 49.26 | 49.18 | 49.01 | 49.51 |
| | Peri-LN | 51.41 | 51.14 | 50.68 | 52.13 |

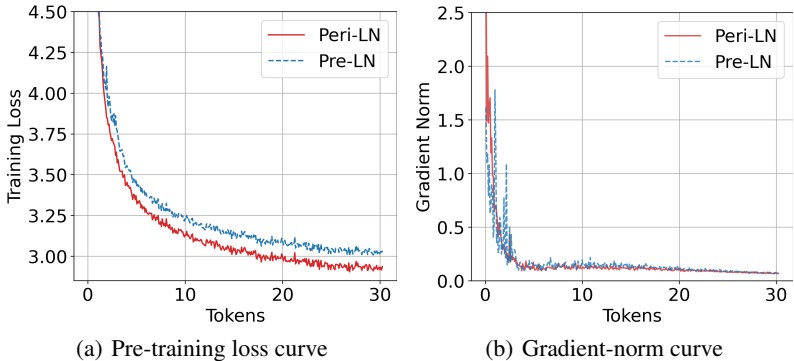

(a) Pre-training loss curve      (b) Gradient-norm curve

*Figure 23.* Comparison of pre-training loss and gradient norms for Pre-LN and Peri-LN architectures with the weight decay coefficient fixed at 0, while all other hyperparameters are held constant.

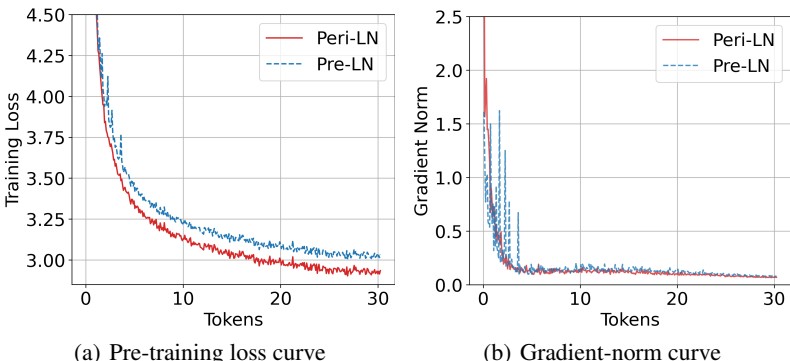

(a) Pre-training loss curve      (b) Gradient-norm curve

*Figure 24.* Comparison of pre-training loss and gradient norms for Pre-LN and Peri-LN architectures with the weight decay coefficient fixed at 0.0033, while all other hyperparameters are held constant.

### G.8.3. WEIGHT INITIALIZATION

We run an additional ablation on weight-initialization schemes. For both Pre-LN and Peri-LN models, we first adopt Xavier initialization (Glorot & Bengio, 2010) and then compare it with He initialization ($2/d$) (He et al., 2015), LeCun initialization

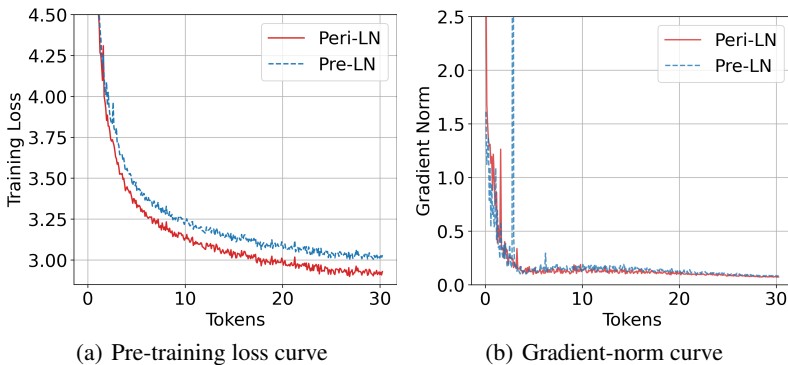

(a) Pre-training loss curve      (b) Gradient-norm curve

*Figure 25.* Comparison of pre-training loss and gradient norms for Pre-LN and Peri-LN architectures with the weight decay coefficient fixed at 0.033, while all other hyperparameters are held constant.

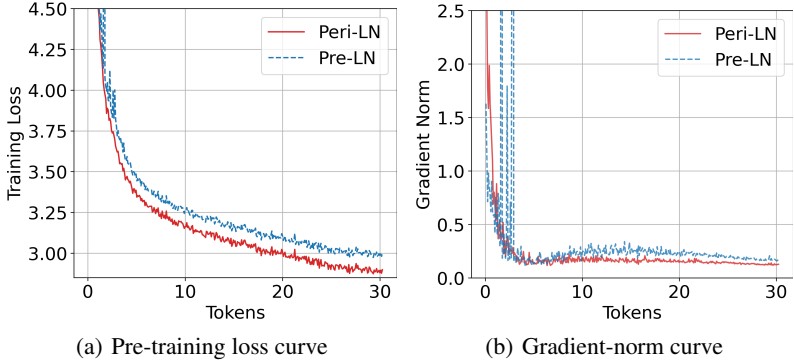

(a) Pre-training loss curve      (b) Gradient-norm curve

*Figure 26.* Comparison of pre-training loss and gradient norms for Pre-LN and Peri-LN architectures with the weight decay coefficient fixed at 0.33, while all other hyperparameters are held constant.

$(1/d)$, and two scaled variants, $10/d$ and $1/(10d)$.

As Table 8 shows, Xavier initialization yields the strongest overall performance, improving on the configurations used in our earlier experiments. Crucially, our central finding remains intact: hidden-state variance sharply grows in Pre-LN Transformers but stays bounded in Peri-LN Transformers. Table 9 confirms the same pattern across all initialization scales, and detailed per-run results appear below.

*Table 8.* Xavier initialization (Glorot & Bengio, 2010) yields better performance compared to our previous weight initialization configurations.

| 400M | Architecture | Baseline(0.02) | Xavier Initialization |
|------|--------------|----------------|----------------------|
| Loss | Pre-LN | 3.03 | 2.95 |
|      | Peri-LN | 2.93 | 2.91 |
| Avg. | Pre-LN | 49.01 | 51.25 |
|      | Peri-LN | 50.68 | 52.04 |

*Table 9.* Effect of weight-initialization variance on final pre-training loss for 400M-parameter Pre-LN and Peri-LN Transformers.

| 400M | Initialization Variance | 10/d | He (2/d) | LeCun (1/d) | 1/(10d) | Baseline (0.02) |
|------|------------------------|------|----------|-------------|---------|-----------------|
| Loss | Pre-LN | 4.526 | 2.965 | 3.005 | 3.012 | 3.035 |
|      | Peri-LN | 3.027 | 2.929 | 2.915 | 2.902 | 2.916 |

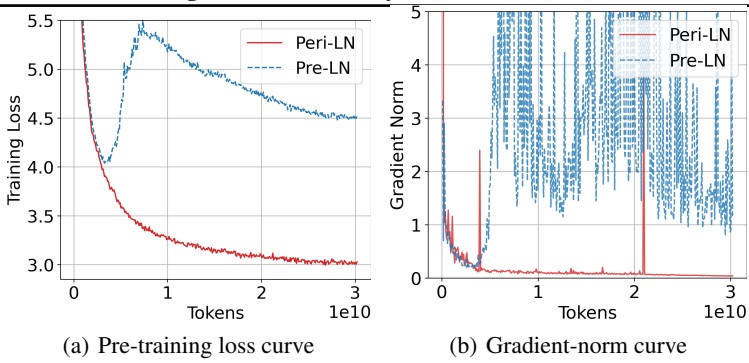

(a) Pre-training loss curve

(b) Gradient-norm curve

*Figure 27.* Comparison of pre-training loss and gradient norms for Pre-LN and Peri-LN architectures with the weight initialization variance set to $10/d$, while all other hyperparameters are held constant.

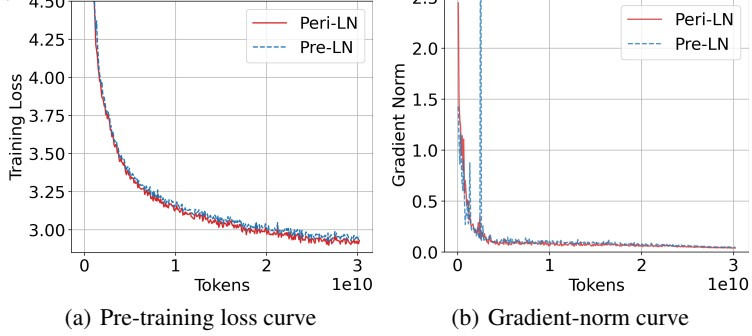

(a) Pre-training loss curve

(b) Gradient-norm curve

*Figure 28.* Comparison of pre-training loss and gradient norms for Pre-LN and Peri-LN architectures with the weight initialization variance set to $2/d$ (He init (He et al., 2015)), while all other hyperparameter are held constant.

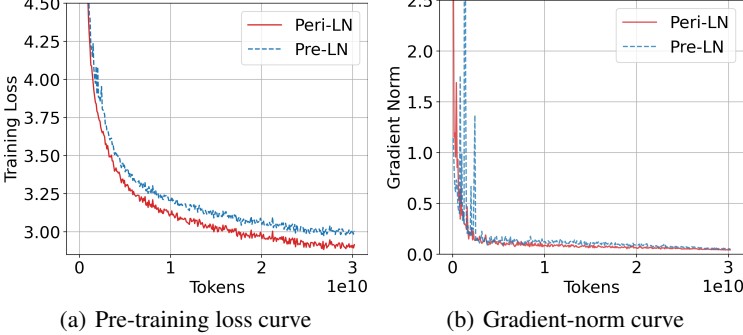

(a) Pre-training loss curve

(b) Gradient-norm curve

*Figure 29.* Comparison of pre-training loss and gradient norms for Pre-LN and Peri-LN architectures with the weight initialization variance set to $1/d$ (LeCun init), while all other hyperparameter are held constant.

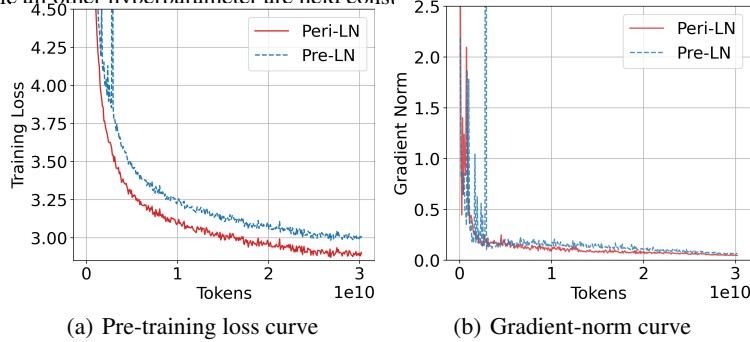

(a) Pre-training loss curve

(b) Gradient-norm curve

*Figure 30.* Comparison of pre-training loss and gradient norms for Pre-LN and Peri-LN architectures with the weight initialization variance set to $1/(10d)$, while all other hyperparameter are held constant.

# H. Output-Layer Normalization with QK-Norm Architecture

Query and Key layer-normalization (QK-Norm) has been widely studied in modern Transformer architectures (Wortsman et al., 2024; Zhai et al., 2023; OLMo et al., 2024). In particular, OLMo et al. (2024) reported that QK-Norm combined with module output layer-normalization (output-LN, $B$ in Figure 31 referred to as "reordered norm" in the OLMo2 paper) improves both training loss and stability. As shown in Figure 31, QK-Norm is applied after the Query and Key projections, similar to output-LN. From another perspective, QK-Norm is also applied immediately before the attention calculation, akin to a Pre-LN approach. In our view, QK-Norm and Pre-LN (placed at $A^2$ and $A$ respectively in Figure 31) serve the same role but differ in certain details. As shown in Figures 32, 33, and 34, the two architectures exhibit comparable performance overall in terms of both training loss and stability.. However, Peri-LN provides a slight performance advantage over the OLMo2-style Peri-LN in the 400M- and 1B-parameter models.

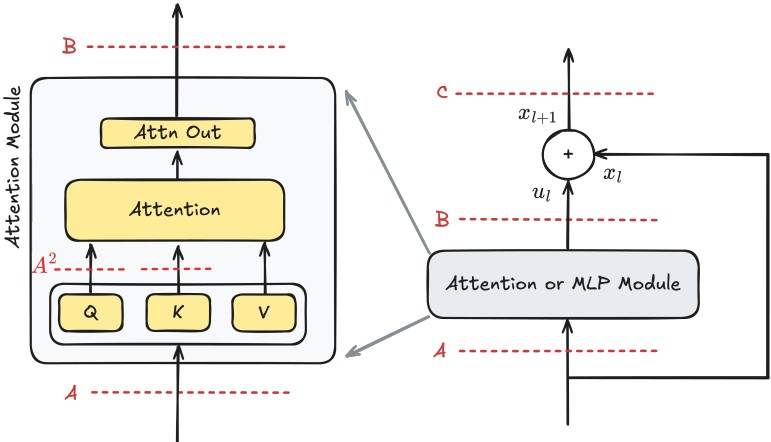

*Figure 31.* QK-layer normalization in the Attention module.

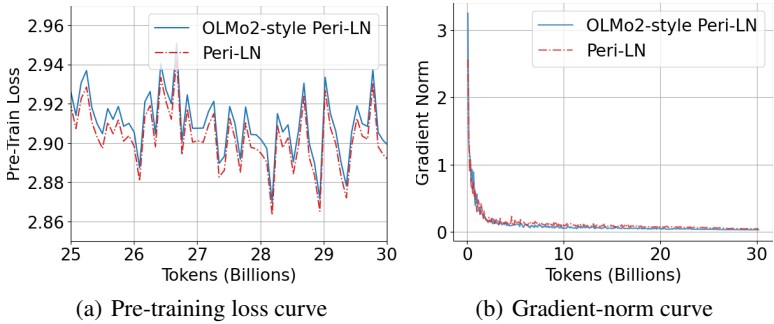

(a) Pre-training loss curve        (b) Gradient-norm curve

*Figure 32.* Comparison of pre-training loss and gradient norm between OLMo2-Style Peri-LN and the Peri-LN architecture. To ensure an accurate comparison, we present the pre-training loss over the final 5B tokens. 400M size model.

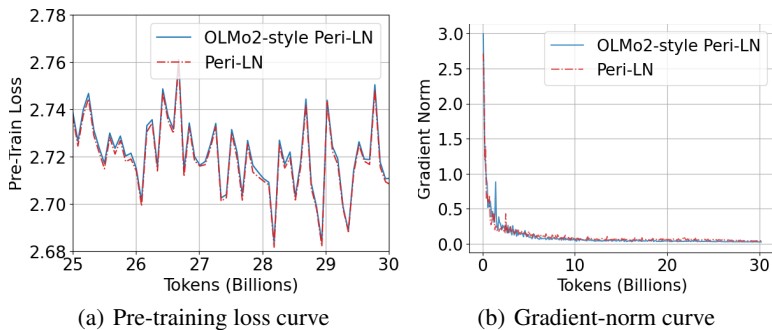

(a) Pre-training loss curve      (b) Gradient-norm curve

*Figure 33.* Comparison of pre-training loss and gradient norm between OLMo2-Style Peri-LN and the Peri-LN architecture. To ensure an accurate comparison, we present the pre-training loss over the final 5B tokens. 1.5B size model.

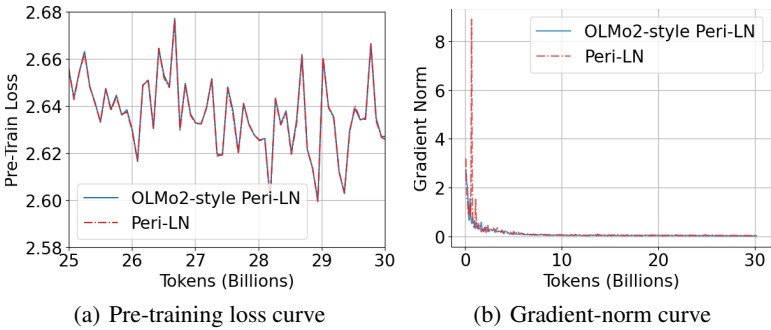

(a) Pre-training loss curve      (b) Gradient-norm curve

*Figure 34.* Comparison of pre-training loss and gradient norm between OLMo2-Style Peri-LN and the Peri-LN architecture. To ensure an accurate comparison, we present the pre-training loss over the final 5B tokens. 3.2B size model.

## I. Training the Transformer using Stochastic Gradient Descent

Using Stochastic Gradient Descent (SGD) for training Transformers is not a common practice. As Zhang et al. (2024) point out, Transformer-based models tend to perform worse with SGD than with Adam by a considerable margin. One reason is that SGD struggles to handle the heterogeneity across different blocks. Although these aspects are certainly intriguing and warrant further investigation, they lie beyond the scope of our current work, as Zhang et al. (2024) also note.

Nonetheless, for someone who might wonder to know, we conduct additional experiments using SGD. We are searching for U-shaped patterns during the learning rate exploration for both Pre-LN & Peri-LN as shown in the Figure 35. We observe that: (1) SGD performs worse than Adam, consistent with findings reported in (Zhang et al., 2024); and (2) Peri-LN demonstrates better performance than Pre-LN. In these results, we use 400 M-parameter Transformers and apply the same configurations as in the main experiments (Section 4.1). We provide detailed training curves in Figures 36, 37, 38.

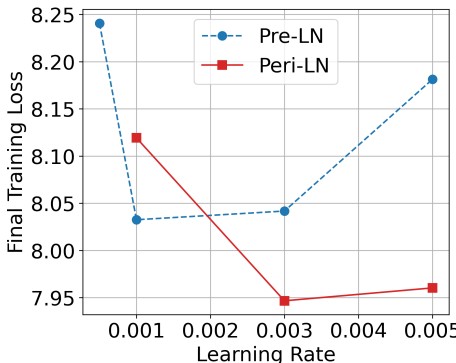

*Figure 35.* Learning Rate Exploration of Pre-& Peri-LN architecture trained with SGD optimizer. The individual training-loss and gradient-norm curves appear in Figures 36, 37, and 38.

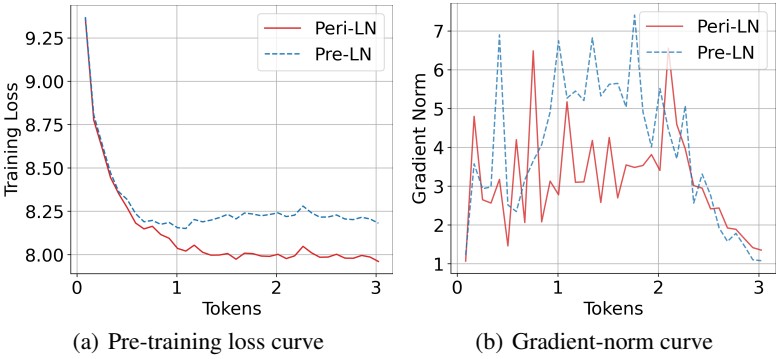

(a) Pre-training loss curve       (b) Gradient-norm curve

*Figure 36.* Using SGD with learning rate $5e - 3$.

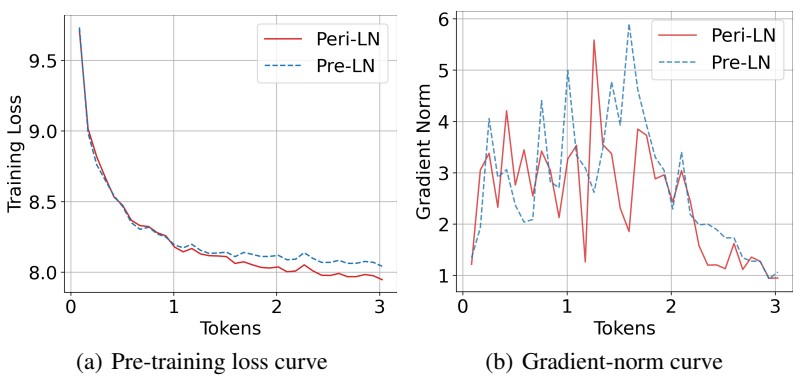

(a) Pre-training loss curve       (b) Gradient-norm curve

*Figure 37.* Using SGD with learning rate $3e - 3$.

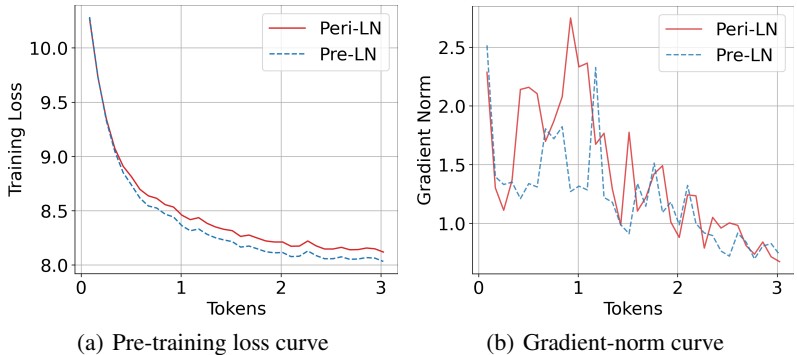

(a) Pre-training loss curve       (b) Gradient-norm curve

*Figure 38.* Using SGD with learning rate $1e - 3$.

# J. Additional Details on Evaluation

## J.1. Detailed Configurations

We utilized the Language Model Evaluation Harness[6]with the HuggingFace Transformers library (Gao et al., 2024; Wolf et al., 2020) to assess overall performance. We employ five different evaluation benchmarks: ARC (Clark et al., 2018), HellaSwag (Zellers et al., 2019), PIQA (Bisk et al., 2020), SIQA (Sap et al., 2019), Winogrande (Sakaguchi et al., 2021). During the pretraining stage, each model was trained under a controlled random seed. We used the training loss at iteration $14,000$—corresponding to the completion of 30B tokens—as our main reference point. When calculating the evaluation score, diverged checkpoints were excluded.

## J.2. Detailed Results on Benchmark Evaluations

In this section, we present the evaluation results for each model trained with five different training seeds. We exclude any diverged scores and average the remaining values, which are then reported in Table 1 in the main text.

### J.2.1. PRE-TRAINING

*Table 10.* Detailed results on pre-training the Peri-LN architecture. These results are averaged to produce the values reported in Table 1. *SEED* denotes pre-training seed.

| Peri-LN | SEED | ARC-Easy | HellaSwag | PIQA | SIQA | Winogrande |
|---|---|---|---|---|---|---|
| | 1 | 0.5758 | 0.3803 | 0.6980 | 0.4115 | 0.5225 |
| | 2 | 0.5728 | 0.3739 | 0.6915 | 0.4017 | 0.5367 |
| 400M | 3 | 0.5842 | 0.3745 | 0.6986 | 0.4125 | 0.5249 |
| | 4 | 0.5800 | 0.3722 | 0.6959 | 0.4038 | 0.5209 |
| | 5 | 0.5627 | 0.3719 | 0.6899 | 0.4028 | 0.5320 |
| | 1 | 0.6599 | 0.4437 | 0.7339 | 0.4304 | 0.5714 |
| | 2 | 0.6591 | 0.4394 | 0.7399 | 0.4145 | 0.5699 |
| 1.5B | 3 | 0.6625 | 0.4357 | 0.7372 | 0.4166 | 0.5627 |
| | 4 | 0.6633 | 0.4367 | 0.7345 | 0.4222 | 0.5667 |
| | 5 | 0.6637 | 0.4416 | 0.7361 | 0.4335 | 0.5612 |
| | 1 | 0.6953 | 0.4734 | 0.7443 | 0.4417 | 0.5872 |
| | 2 | 0.6839 | 0.4684 | 0.7427 | 0.4324 | 0.6054 |
| 3.2B | 3 | 0.6902 | 0.4680 | 0.7486 | 0.4243 | 0.5967 |
| | 4 | 0.6864 | 0.4700 | 0.7427 | 0.4273 | 0.5935 |
| | 5 | 0.6806 | 0.4698 | 0.7372 | 0.4243 | 0.6054 |

*Table 11.* Detailed results on pre-training the Pre-LN architecture. These results are averaged to produce the values reported in Table 1. *SEED* denotes pre-training seed.

| Pre-LN | SEED | ARC-Easy | HellaSwag | PIQA | SIQA | Winogrande |
|---|---|---|---|---|---|---|
| | 1 | 0.5669 | 0.3609 | 0.7008 | 0.4002 | 0.5359 |
| | 2 | | | Diverged | | |
| 400M | 3 | 0.5354 | 0.3328 | 0.6741 | 0.3905 | 0.4957 |
| | 4 | | | Diverged | | |
| | 5 | 0.5438 | 0.3314 | 0.6888 | 0.4012 | 0.4949 |
| | 1 | 0.6326 | 0.4259 | 0.7242 | 0.4263 | 0.5691 |
| | 2 | 0.6019 | 0.3924 | 0.7111 | 0.3992 | 0.5627 |
| 1.5B | 3 | 0.6077 | 0.3932 | 0.7008 | 0.4125 | 0.5272 |
| | 4 | 0.6111 | 0.3886 | 0.7187 | 0.4135 | 0.5225 |
| | 5 | 0.6221 | 0.3941 | 0.7160 | 0.4099 | 0.5438 |
| | 1 | 0.6688 | 0.4588 | 0.7470 | 0.4273 | 0.5919 |
| | 2 | | | Diverged | | |
| 3.2B | 3 | | | Diverged | | |
| | 4 | 0.6359 | 0.4259 | 0.7301 | 0.4263 | 0.5564 |
| | 5 | | | Diverged | | |

---

[6]https://github.com/EleutherAI/lm-evaluation-harness

*Table 12.* Detailed results on pre-training the Post-LN architecture. These results are averaged to produce the values reported in Table 1. *SEED* denotes pre-training seed.

| Post-LN | SEED | ARC-Easy | HellaSwag | PIQA | SIQA | Winogrande |
|---|---|---|---|---|---|---|
| | 1 | 0.3413 | 0.2881 | 0.6311 | 0.3378 | 0.5067 |
| | 2 | 0.3691 | 0.2886 | 0.6132 | 0.3337 | 0.5099 |
| 400M | 3 | 0.3632 | 0.2889 | 0.6257 | 0.3603 | 0.5051 |
| | 4 | 0.3603 | 0.2920 | 0.6262 | 0.3490 | 0.5012 |
| | 5 | 0.3510 | 0.2880 | 0.6170 | 0.3434 | 0.5209 |
| | 1 | 0.4268 | 0.3121 | 0.6659 | 0.3628 | 0.5185 |
| | 2 | 0.4196 | 0.3150 | 0.6654 | 0.3639 | 0.5004 |
| 1.5B | 3 | | | Diverged | | |
| | 4 | 0.4285 | 0.3212 | 0.6730 | 0.3511 | 0.4775 |
| | 5 | 0.4419 | 0.3193 | 0.6643 | 0.3557 | 0.5154 |
| | 1 | 0.4731 | 0.3427 | 0.6774 | 0.3664 | 0.5343 |
| | 2 | 0.4638 | 0.3326 | 0.6779 | 0.3577 | 0.4917 |
| 3.2B | 3 | 0.3956 | 0.3321 | 0.6143 | 0.3408 | 0.5067 |
| | 4 | 0.4663 | 0.3380 | 0.6692 | 0.3685 | 0.5178 |
| | 5 | 0.4663 | 0.3340 | 0.6839 | 0.3577 | 0.5043 |

## J.2.2. SUPERVISED FINE-TUNING

*Table 13.* Detailed results on SFT with Peri-LN architecture. These results are averaged to produce the values reported in Table 1. *SEED* denotes pre-training seed.

| Peri-LN | SEED | ARC-Easy | HellaSwag | PIQA | SIQA | Winogrande |
|---|---|---|---|---|---|---|
| | 1 | 0.5800 | 0.3819 | 0.6991 | 0.4145 | 0.5328 |
| | 2 | 0.5783 | 0.3783 | 0.6921 | 0.4038 | 0.5391 |
| 400M | 3 | 0.5888 | 0.3806 | 0.6980 | 0.4222 | 0.5288 |
| | 4 | 0.5892 | 0.3738 | 0.6948 | 0.4089 | 0.5099 |
| | 5 | 0.5783 | 0.3757 | 0.6991 | 0.4099 | 0.5312 |
| | 1 | 0.6633 | 0.4502 | 0.7356 | 0.4304 | 0.5746 |
| | 2 | 0.6641 | 0.4437 | 0.7405 | 0.4207 | 0.5706 |
| 1.5B | 3 | 0.6671 | 0.4454 | 0.7454 | 0.4207 | 0.5620 |
| | 4 | 0.6700 | 0.4455 | 0.7378 | 0.4284 | 0.5659 |
| | 5 | 0.6688 | 0.4478 | 0.7421 | 0.4324 | 0.5620 |
| | 1 | 0.7058 | 0.4810 | 0.7486 | 0.4422 | 0.5880 |
| | 2 | 0.6898 | 0.4774 | 0.7437 | 0.4391 | 0.6054 |
| 3.2B | 3 | 0.6995 | 0.4770 | 0.7481 | 0.4278 | 0.5912 |
| | 4 | 0.6911 | 0.4777 | 0.7432 | 0.4350 | 0.5943 |
| | 5 | 0.6894 | 0.4781 | 0.7448 | 0.4319 | 0.6046 |

*Table 14.* Detailed results on SFT with Pre-LN architecture. These results are averaged to produce the values reported in Table 1. *SEED* denotes pre-training seed.

| Pre-LN | SEED | ARC-Easy | HellaSwag | PIQA | SIQA | Winogrande |
|--------|------|----------|-----------|------|------|------------|
| | 1 | 0.5762 | 0.3625 | 0.7078 | 0.4058 | 0.5343 |
| | 2 | | | N/A | | |
| 400M | 3 | 0.5370 | 0.3339 | 0.6757 | 0.3905 | 0.4972 |
| | 4 | | | N/A | | |
| | 5 | 0.5509 | 0.3372 | 0.6893 | 0.4074 | 0.4886 |
| | 1 | 0.6385 | 0.4310 | 0.7247 | 0.4227 | 0.5620 |
| | 2 | 0.6035 | 0.3934 | 0.7095 | 0.4038 | 0.5572 |
| 1.5B | 3 | 0.6098 | 0.3944 | 0.7035 | 0.4150 | 0.5257 |
| | 4 | 0.6208 | 0.3929 | 0.7182 | 0.4161 | 0.5272 |
| | 5 | 0.6258 | 0.4017 | 0.7171 | 0.4181 | 0.5391 |
| | 1 | 0.6785 | 0.4681 | 0.7568 | 0.4345 | 0.5825 |
| | 2 | | | N/A | | |
| 3.2B | 3 | | | N/A | | |
| | 4 | 0.6427 | 0.4293 | 0.7274 | 0.4299 | 0.5580 |
| | 5 | | | N/A | | |

*Table 15.* Detailed results on SFT with Post-LN architecture. These results are averaged to produce the values reported in Table 1. *SEED* denotes pre-training seed.

| Post-LN | SEED | ARC-Easy | HellaSwag | PIQA | SIQA | Winogrande |
|---------|------|----------|-----------|------|------|------------|
| | 1 | 0.4428 | 0.3307 | 0.6583 | 0.3797 | 0.5099 |
| | 2 | 0.4280 | 0.3208 | 0.6404 | 0.3746 | 0.5178 |
| 400M | 3 | 0.4693 | 0.3241 | 0.6578 | 0.3905 | 0.5122 |
| | 4 | 0.4680 | 0.3247 | 0.6610 | 0.3726 | 0.4830 |
| | 5 | 0.4520 | 0.3283 | 0.6572 | 0.3849 | 0.5225 |
| | 1 | 0.5316 | 0.3774 | 0.6980 | 0.3889 | 0.5359 |
| | 2 | 0.4731 | 0.3316 | 0.6719 | 0.3813 | 0.5028 |
| 1.5B | 3 | | | N/A | | |
| | 4 | 0.5387 | 0.3546 | 0.6779 | 0.3864 | 0.4909 |
| | 5 | 0.5261 | 0.3510 | 0.6752 | 0.3767 | 0.5209 |
| | 1 | 0.5623 | 0.4029 | 0.7008 | 0.3920 | 0.5051 |
| | 2 | 0.5417 | 0.3644 | 0.6823 | 0.3833 | 0.5264 |
| 3.2B | 3 | 0.4444 | 0.3604 | 0.6333 | 0.3618 | 0.5043 |
| | 4 | 0.5400 | 0.3645 | 0.6844 | 0.3823 | 0.5020 |
| | 5 | 0.5341 | 0.3677 | 0.6942 | 0.3976 | 0.5012 |

## K. Additional Discussions of Precision Constraints Imposed by Pre-LN

This section provide additional discussions of Section 7.2. Given that both Pre-LN and Peri-LN exhibit a structural property whereby large values do not readily disappear once they arise, it is important to monitor the occurrence of these extreme activations. Pre-LN's additive residual path often produces activations that approach, and occasionally exceed, the FP16 (float16) representable maximum[7]. To quantify how often these values would overflow FP16 yet remain within the BF16 (bfloat16) range[8], we measure the 100 largest absolute hidden-state values (the global top-100 tokens) for each Pre-LN and Peri-LN 3.2B-parameter Transformer. The shaded region indicates the range of these global top-100 tokens. The blue curve (with shaded band) represents a Pre-LN model, the red curve a Peri-LN model, and the dashed orange line denotes the FP16 representable maximum.

As shown in Figure 11, consistent with the observations of Sun et al. (2024), activations in the Pre-LN model routinely exceed the FP16 bound from the very first 0.5B training tokens, with the overshoot becoming more pronounced in deeper layers—an indication of growing numerical instability. By contrast, Peri-LN consistently maintains a comfortable margin below the FP16 limit throughout training. This finding underscores that the choice between FP16 and BF16 is not merely a hardware preference; it is tightly coupled to how hidden-state magnitudes evolve within the network.

Since NVIDIA V100 GPUs do not support BF16, these results imply that training and inference with Pre-LN models on such hardware are inherently disadvantaged. Moreover, from the standpoint of large-language-model quantization (Dettmers et al., 2022; Lee et al., 2023; Kim et al., 2023; Heo et al., 2024; Lee et al., 2025), the prevalence of massive activations in Pre-LN can severely disrupt outlier-aware compression algorithms. When aggressive low-precision compression is the goal, the Pre-LN architecture's tendency to generate extreme hidden state values therefore constitutes a particularly challenging obstacle.

Meta AI's publicly released OPT training logbooks and chronicles document repeated episodes of gradient divergence and loss spikes encountered while pre-training entirely in FP16 precision [9]. Since FP16 saturates at an absolute value of $65,504$, any hidden state activation that exceeds this threshold silently overflows, corrupting the forward pass and, through back-propagation, inducing erratic gradients. Earlier analyses of OPT (Zhang et al., 2022) therefore ascribe much of the observed instability to numerical overflow, a view formalized in our Proposition 3.1, which shows how such out-of-range activations propagate into severe gradient pathologies. These external reports provide further corroboration that architectures prone to generating large-magnitude activations—as Pre-LN does—require either a wider numerical format (e.g., BF16) or explicit magnitude-regularization to ensure stable large-scale training.

---

[7]https://en.wikipedia.org/wiki/Bfloat16_floating-point_format
[8]https://en.wikipedia.org/wiki/Half-precision_floating-point_format
[9]https://github.com/facebookresearch/metaseq/blob/main/projects/OPT/chronicles/10_percent_update.md

## L. Additional Discussions of Hidden State Representations

As shown in Figure 39(a), Post-LN exhibits smaller angular distances due to the LN being located on the main path, whereas Pre-LN and Peri-LN begin with very similar states. As shown in Figure 39(c), at the end of training, Pre-LN tends to produce more redundant hidden state representations compared to the others. This phenomenon may stem from Pre-LN's repeated residual additions, which amplify certain representations over others. We use 30B tokens trained 400M size model in this experiments. For dataset, we utilize 256 random samples of RedPajama-Data-1T (Computer, 2023) for this results.

To investigate further, we focus on module-output normalization, the primary factor distinguishing Pre-LN from Peri-LN. As shown in Figure 39(b), the learnable scale starts around 1 in the early stages of training and gradually changes with increasing depth. Because Peri-LN preserves the identity path, it appears to adjust the scale of the module output accordingly. This suggests that the exponential growth of the main path's magnitude in Pre-LN diminishes the relative contribution of individual modules, resulting in more redundant hidden representations. Figure 39(d) shows that fixing the learnable scale of Peri-LN's module output LN at 1 causes the main path contribution to decrease in deeper layers. This finding confirms the role of module output normalization in controlling hidden state redundancy.

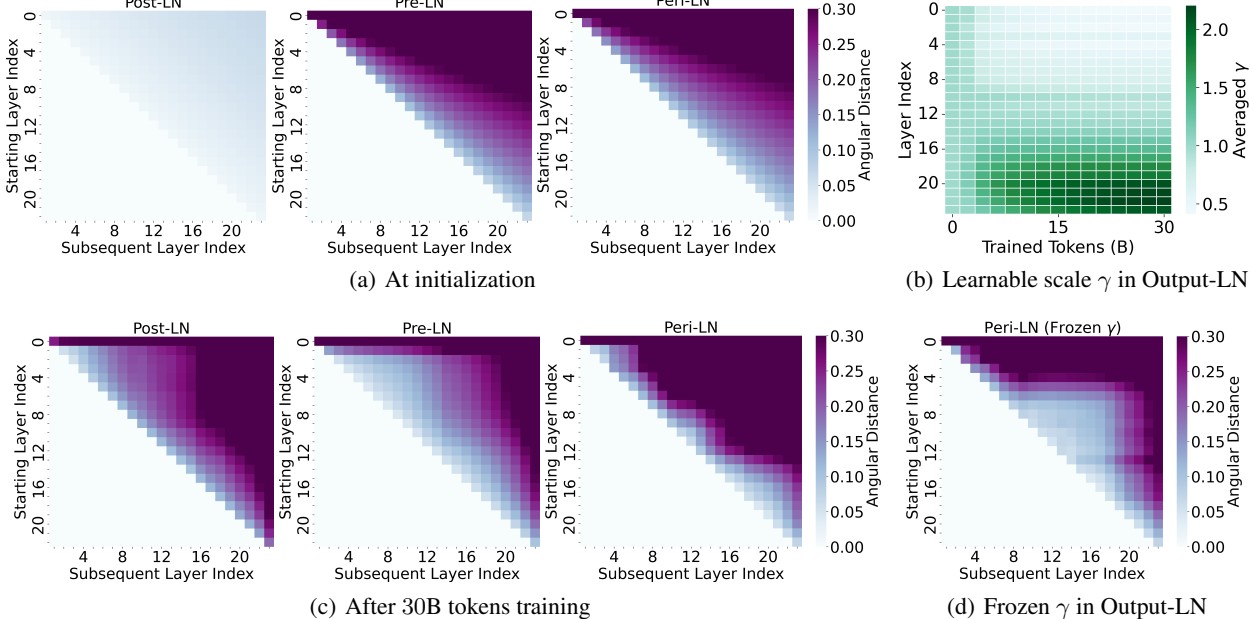

*Figure 39.* Angular distance of hidden state is presented in Figure 39(a), 39(c), and 39(d). In Figure 39(b), we monitor $\gamma$ of every Output-LN in Peri-LN during training. We use 30B tokens trained 400M size model in this experiments.

