# OpenReview forum: "Peri-LN: Revisiting Normalization Layer in the Transformer Architecture"
_ICML.cc/2025/Conference — ICML 2025 poster_

### Official Review · Reviewer_Rrms · 2025-03-09

**Overall Recommendation:** 2

**Summary:**

Extending existing works on the placement of layer normalization in transformers, such as Pre-LN and Post-LN, this study proposes specific positions to apply layer normalizations, called Peri-LN configurations. The authors claim improved theoretical properties such as variance accumulation and gradient behavior, which ensures advantages from existing placements. Experiments validate the improved performance from the Peri-LN.

**Claims And Evidence:**

I think there are several problems in the theoretical parts. See the “Theoretical Claims” below.

**Essential References Not Discussed:**

N/A

**Experimental Designs Or Analyses:**

I think the amount of experiments looks adequate to investigate the validity of the proposed method.

**Methods And Evaluation Criteria:**

Yes, the experiments look well-designed, and the authors provided extensive results.

**Other Comments Or Suggestions:**

N/A

**Other Strengths And Weaknesses:**

Other strengths of the proposed method would be that it is easy to deploy in practical source code by injecting few lines of code.

**Questions For Authors:**

See “Theoretical Claims” above.

**Relation To Broader Scientific Literature:**

The study on foundation models would advance general machine learning fields and further broader scientific literature.

**Theoretical Claims:**

- In Appendix D.1, the term $o$, which is the output of MLP and skip connection, would be a feature in a layer module. However, the authors apply softmax to $o$ to obtain $p$ for their proof, which looks significantly unnatural. The authors should clarify whether $o$ is an element of a layer or whether this proof is intended to narrow down to the scenario of the last layer.
- The proof in Appendix D only considers MLP, and there is no theoretical proof for MHSA.
- In the proof in Appendix D, when using the chain rule, I think summation is necessary in multivariates. Please check this point.
- In Eq. 29 in Appendix D, the term $\gamma$ is assumed to be positive. Although $\gamma$ is commonly initialized to one, it frequently becomes negative during the training. I think this part is not strictly correct either.
- The proof in Appendix D was performed using RMSNorm instead of layer normalization and ReLU instead of GELU. These approximations might be necessary for simplicity in theoretical proofs, but adopting them may give the impression that the proof is not perfect.
- Overall, I evaluate the theoretical part of this manuscript as not enough for publication in ICML. The authors have omitted important parts in the actual proof compared to the wide claim range, and several parts of the proof look incorrect.

---

> ### Author Rebuttal · Authors · 2025-03-31
>
> ### **1) “$o$ seems intermediate; softmax is applied in an unnatural way.”**
> We believe there may be a misunderstanding regarding the reviewer’s concern that $o$ is an intermediate representation and that softmax is applied in an unnatural manner. As noted in Section 3.4 and Section D, our theoretical analysis focuses on the final MLP layer. This choice is motivated by two reasons: (1) the gradient norm at the final layer is empirically known to be the most unstable (see Figure1); and (2) it allows for a rigorous mathematical analysis without requiring approximations or assumptions. Analyzing the final layer for theoretical insight is an established practice in the literature (see Theorem 1 in [1]).
> Importantly, our empirical observations show that other layers exhibit similar trends in gradient behavior and hidden-state variance (see Section 4.4), supporting the generality of our theoretical insights.
>
> [1] Xiong et al. "On layer normalization in the transformer architecture." ICML 2020.
>
> ---
> ### **2) “No theoretical proof for MHSA”**
> As discussed in our Response #1, we focus on the final MLP sub-layer rather than the intermediate MHSA layers. For these reasons, we introduce a proposition centered on the MLP layer, aiming to explain the phenomena observed in our experiments.
>
> ---
> ### **3) “Summation is necessary in multivariates”**
> We respectfully note that the comment regarding explicit summation in multivariate expressions appears to stem from a misunderstanding: in our matrix notation, the summation is inherently handled by matrix multiplication. In componentwise notation, the multivariate chain rule involves a summation over the relevant indices. However, in matrix notation, the summation is taken care of by matrix multiplication. Please refer to Theorem 5.3 in [2].
>
> [2] Colley, Susan Jane. Vector calculus. PEARSON EDUCATION LIMITED, 2012.
>
> ---
> ### **4) “$\gamma$ is assumed positive, yet it can become negative in practice.”**
> We appreciate this point. In theory, $\gamma$ is a scaling parameter primarily intended to adjust magnitude, so the mathematical derivation naturally assumes $\gamma > 0$. In this paper, we assume $\gamma$ remains positive for theoretical simplicity, and we will include a clarification of this assumption in the revised manuscript. Empirically, we further verified $\gamma$ during by monitoring the magnitude across all checkpoints for a 30B-token run; it stayed *strictly positive* in every layer. We would like to emphasize that, as discussed in Section 4.4.1, even when we freeze $\gamma$ to 1, Peri-LN still retains its main benefits.
>
> ---
> ### **5) “RMSNorm <-> LN or ReLU <-> GeLU might not match exactly”**
> We understand the concern that RMSNorm and ReLU may feel “approximate” compared to LN and GeLU. Our motivation was analytical tractability:
> - RMSNorm omits mean-centering, which simplifies the Jacobian while still capturing the main effect of normalization (rescaling by the vector’s norm). LayerNorm introduces an additional term subtracting the mean, but in large dimensions, mean removal has a smaller relative effect—so the bounding principle remains largely the same [3].
> - ReLU is piecewise-linear, making it straightforward to compute partial derivatives explicitly. Since we are only considering the gradient with respect to $W^{(2)}$ in the final layer, it makes no difference whether the preceding hidden activation function is ReLU or GeLU. In either case, the hidden activation $h$ is treated as a constant when differentiating with respect to $W^{(2)}$, so the conclusions regarding stability with respect to $\||h\||$ remain the same.
>
> Hence, these substitutions allow us to *cleanly* show how placing LN at different points can dampen or amplify the backpropagated gradients. We agree it is not a full 1:1 equivalence to LN or GeLU, but the essential mechanics are preserved. In the revised text, we will add disclaimers that our derivation is an instructive idealization—RMSNorm vs. LayerNorm and ReLU vs. GeLU do not qualitatively change the conclusion about how LN placement impacts model dynamics.
>
> [3] Zhang et al. "Root mean square layer normalization." Neurips 2019.
>
> ---
> ### **Concluding Remarks**
> It seems there has been a misunderstanding, and we kindly ask you to reevaluate our work in light of its theoretical and experimental contributions. As Reviewer 1xqQ and AtH3 highlighted, we do believe that our theoretical contribution is solid. Your comments have prompted us to refine our exposition and add more detailed supporting materials. While we respectfully maintain that our core derivations—even with simplified assumptions (RMSNorm instead of LN, ReLU instead of GeLU)—provide a valid perspective on how LN placement influences Transformers, we will include additional clarifications and empirical validations in a revised manuscript. We hope these efforts will address your concerns and more clearly convey the findings of our work. Thank you again for your time.

---

> > ### Comment · Reviewer_Rrms · 2025-04-03
> >
> > Thank you for your response and clarification, and now I understand my misunderstanding; the proof targets the last weight, not the whole layers. Then, there is no problem with (1). Thank you for checking (3), and I understood that there is no problem with it. I checked this manuscript again, but then, I think the theoretical claims of the authors rather become narrow. As the authors claim about the placement of LN in the sub-layer, I naturally misunderstood that the proof would analyze arbitrary intermediate layers; though the authors consider the placement of LN for the sub-layer of the transformer, the theoretical analysis narrows down to a very special case compared with the generic coverage of the claim. The logical flow up to Section 3.3 discusses an arbitrary layer, but Section 3.4 now discusses the final layer. I think there should be sufficient transition comments for targeting the last layer right here for analysis to prevent other readers from similar misunderstandings.
> >
> > Thus, my thoughts (2, 4, 5) that there are many leaps and gaps between the authors' proposed method and Proposition 3.1 have not changed. They only target the theoretical properties of the last layer, not all layers, do not analyze MHSA and only prove it for MLP, approximate LayerNorm into RMSNorm and GELU into ReLU, and assume \gamma to be positive. I think these implicit assumptions should have been listed more precisely as “Assumptions” before presenting the Proposition. I understand that the authors have their own convincing logical basis for those assumptions as provided in their response, but they should have been sufficiently mentioned in the main text rather than discussing it here.
> >
> > I hope that the underlying assumptions will be clearly presented in the manuscript. I raised the score from 1 to 2, assuming that the manuscript will be revised to mention all of them, but I still think that there is a large gap between the coverage of authors' claims and the theoretical analysis, which is the reason that I evaluate this manuscript below the borderline.

---

> > > ### Author Response · Authors · 2025-04-04
> > >
> > > **Response to Reviewer Rrms**
> > >
> > > We sincerely appreciate the time you have taken to revisit your initial assessment and for raising your score from 1 to 2. We understand your concerns regarding the scope of our theoretical analysis and would like to provide additional clarifications that may further illuminate our rationale and encourage a more positive overall assessment.
> > >
> > > ---
> > > ### **Revisiting the Focus of Our Theoretical Analysis**
> > > 1. **Why the Final Layer?**
> > >    In large-scale Transformer training, the final layer is empirically known to exhibit the most pronounced gradient instability. Consequently, Section 3.4 focuses the theoretical lens to this final MLP layer. This choice is motivated by the desire to give a mathematically rigorous and tractable argument at the point in the network where instability is most critical. By isolating the final layer, we can reduce additional confounding factors and avoid introducing yet more assumptions or approximations.
> > > 2. **Bridging Theory and Practice**
> > > Although our theoretical discussion centers on the final layer, our empirical findings in Sections 4.3 and 4.4 confirm that similar trends in gradient behavior appear throughout multiple sub-layers. In other words, **we do not rely on final-layer theory alone** to justify Peri-LN. Rather, we use it in tandem with **extensive experimental validation** to connect theoretical insights about LN placement to real-world training outcomes. By combining a mathematically rigorous final-layer analysis with comprehensive experiments, we aim to strengthen our overall argument without resorting to additional approximations that might compromise theoretical clarity.
> > >
> > > 3. **Assumptions and Simplifications**
> > >    We agree that approximating LN with RMSNorm and GELU with ReLU, as well as assuming $\gamma>0$, should have been listed more explicitly as assumptions in the main text, rather than mentioned in the appendix or rebuttal. We accept your advice and plan to:
> > >    - Introduce a concise subsection outlining these assumptions prior to Proposition 3.1,
> > >    - Emphasize that carefully choosing the final MLP layer allows us to remain mathematically rigorous while keeping the derivations tractable, and
> > >    - Show complementary experiments indicating that these simplifications do not significantly change the qualitative conclusions about LN placement.
> > >
> > > ---
> > > ### **Motivation for Studying Peri-LN**
> > >
> > > Beyond the theoretical analysis, we would like to reiterate why Peri-LN warrants close attention:
> > >
> > > - Empirical Adoption but Limited Explanation: Several major open-source models (e.g., Olmo2, Gemma2, Gemma3) already employ a Peri-LN–like structure However, *prior technical reports have not discussed what makes such a design beneficial* in contrast to widely studied Pre-LN or Post-LN. By investigating Peri-LN in detail, we hope to highlight the structural advantages responsible for its observed success in these implementations.
> > >
> > >
> > > - Comparative Analysis: While Pre-LN and Post-LN have been extensively studied, the “Peri-LN” approach *remains relatively unexplored despite being adopted in practice*. We aim to fill that gap by providing both empirical and theoretical perspectives on why Peri-LN helps stabilize large-scale Transformer training, mitigate activation spikes, and yield robust convergence.
> > >
> > > - Practical & Structural Insights: As large language models (LLMs) become ever more crucial across domains, subtle differences in LN placement can affect training stability, final performance, and computational resource requirements. We believe that a *combination of thorough experimentation and targeted theoretical exploration* is critical for understanding these architectural choices in depth.
> > >
> > > ---
> > > ### **Final Remarks**
> > > Our intent is to offer a holistic analysis that is both mathematically rigorous and empirically validated. We hope these refinements—together with our extensive experiments—convince you to consider a more favorable view of the manuscript’s overall contribution.
> > > If there are additional points you would like us to address, or if you have further questions about bridging theory and practice, we would be delighted to discuss them. Thank you again for your time and for raising your score, and we hope our explanations have provided useful context for the scope and intentions of our work.

---

### Official Review · Reviewer_1xqQ · 2025-03-13

**Overall Recommendation:** 2

**Summary:**

This paper focuses on how different LN strategies influence on training dynamics
in transformer architectures training and present a LN strategy called Peri-LN ,
which applies LN around the sub-layer.

By theoretical analysis and experiments, the authors suggest that Peri-LN can
not only improves gradient stability and final loss but also plays a critical role in
reducing hidden-state redundancy, which shows better performance than Post-LN and Pre-LN.

**Claims And Evidence:**

The claims are enlightening, which is supported by both theoretical and
experimental evidence.

The analysis and results are comprehensive, which offers a through
comparison of Post-LN, Pre-LN and Peri-LN.

**Essential References Not Discussed:**

No.

**Experimental Designs Or Analyses:**

The experiments involve large scale experiments on multiple experiment
settings . The authors compare the performance of different LN strategies with
separate benchmarks and systematically analyse the mechanics of Peri-LN
from different perspective. The analyses are comprehensive.

My concern is about the initilization method of the network, which may affect the conclusion of this paper. Please see **Questions For Authors** for details.

**Methods And Evaluation Criteria:**

The methods and evaluation criteria are reasonable.

However, the experiments only evaluate the performance on language
benchmarks. Assessing the effectiveness on other tasks would provide a more
comprehensive understanding of the utility of different LN strategies.

**Other Comments Or Suggestions:**

No.

**Other Strengths And Weaknesses:**

No.

**Questions For Authors:**

Q1: Will the authors consider to provide further theoretical analysis, for example
changing MLP into attention?

Q2: Could the advantage of Peri-LN preserve when it transfer to other tasks, for
example vision or Multimodal tasks?

Q3: We know that LN can control the distribution of hidden neurons, but can not control the gradient strictly. Therefore, **Initialization Methods** are still essential in training a DNN. Which initialization method did the authors apply? He initialization (with variance 2/d), LeCun initialization (with variance 1/d) or others? Could the authors provide the results under different initialization methods? For example, change the variance to 10/d or 1/10d, which may address the issues of gradient vanishing or gradient exploding. I am curious about the results under these initializations, although they are not common in current trainings.

**Relation To Broader Scientific Literature:**

The key contribution is a in-depth analysis of different LN strategies in largescale transformer structures. The author summarize a new LN strategy termed
Peri-LN and bring a perspective to how we apply normalization technologies in
application.

**Theoretical Claims:**

No errors in theoretical claims are found. This paper have solid theoretical
analysis, which provide convincing evidence of the conclusions.

However, the theoretical analysis in Proposition 3 employs MLP for the analysis,
which is different from the attention module used in the actual transformer
architectures.

---

> ### Author Rebuttal · Authors · 2025-03-31
>
> ### **1) Extending Analysis to Other Layers**
> Thank you for highlighting this. As noted in Section 3.4 and Section D, our analysis focuses on the final layer. Following Theorem 1 in [1], we analyze the last layer because its gradients are often the largest in magnitude. We chose $W^{(2)}$ (the final linear projection in the MLP) as a representative example since it most directly feeds into the residual connection $(x + a)$, making it more transparent to illustrate how gradient norms can explode or vanish. This direct link to the residual path can significantly impact gradient stability in subsequent layers. Nonetheless, for a more comprehensive understanding of LLMs training dynamics, extending this theoretical foundation to other components (such as attention) is indeed important. We appreciate the reviewer’s insight on this matter and plan to pursue this direction as part of our future research.
>
> [1] Xiong, Ruibin, et al. "On layer normalization in the transformer architecture." ICML 2020.
>
> ---
>
> ### **2) Extending Exploration to Vision or Multimodal Tasks**
> Due to time and resource constraints, we could not run additional experiments in vision or multimodal settings. However, existing literature lends some support to the broader applicability of our findings. For instance, Sun et al. [2] reports that in ViT (Vision Transformer) architectures, massive activations can also emerge under a Pre-LN setup, paralleling what we observe in language models. This similarity suggests that the insights from our Peri-LN analysis could extend beyond pure language tasks. Given the trend of integrating LLMs into large-scale vision-language models, the reviewer’s question is indeed highly relevant. As outlined in our paper, we focused on large language models as the primary use case for Peri-LN. However, we see great potential in exploring vision or multimodal tasks in future work, building on the theoretical and empirical observations presented here.
>
> [2] Sun, Mingjie, et al. "Massive activations in large language models." COLM 2024.
>
> ---
>
> ### **3) Weight Initialization: Additional Experiments & Clarification**
> - **Additional Experiments**: In response to the reviewer’s question, we conducted additional experiments to explore different weight initialization methods. In this study, for both Pre-LN and Peri-LN architectures, we apply Xavier initialization [3]. As shown in the table below, Xavier initialization yields better performance compared to our previous weight initialization configurations. We also confirm that our main observation still holds: *large variance occurs in Pre-LN Transformers but not in Peri-LN Transformers*. We will provide detailed results that gradient and loss spikes still occur in Pre-LN training curves. Thank you for your insightful guidance on improving the experimental quality of the paper.
> - **Experimental Settings**: We pre-train the 400M-parameter Transformers on 30B tokens each under the controlled same training seed. We measure the training loss and averaged benchmark score for these experiments under the same evaluation settings used in Table 2 of the paper. Other configurations follow those outlined in Section 4.1.
> - **Clarification on the Weight Initialization**: We acknowledge that we did not provide sufficient detail about initialization methods in the original manuscript. In the experiments discussed in the paper, we initialized the weights using a zero-mean Gaussian distribution with a standard deviation of 0.02. We will clarify these details in the revised manuscript.
>
> |400M|Architecture|Paper|Xavier Initialization [3]|
> |-|-|-|-|
> |Loss|Pre-LN|3.03|2.95|
> | |Peri-LN|2.93|2.91|
> |Avg.|Pre-LN|49.01|51.25|
> | |Peri-LN|50.68|52.04|
>
> [3] Glorot, Xavier, and Yoshua Bengio. "Understanding the difficulty of training deep feedforward neural networks." international conference on artificial intelligence and statistics. JMLR Workshop and Conference Proceedings, 2010.
>
> ---
>
> ### **Concluding Remarks**
> Once again, we sincerely appreciate your insightful feedback. Your questions on extending Peri-LN’s theoretical analysis to other components and applying it to tasks beyond language have highlighted valuable directions for our future work. We are committed to further investigating these avenues—particularly how Peri-LN might generalize to vision or multimodal settings—and to incorporating additional details on initialization strategies to ensure that our results remain transparent and consistent. We hope these efforts will address your questions and more clearly convey the findings of our work. We will make sure to incorporate your valuable suggestions into the revised manuscript. If you have any further questions or topics you would like to discuss, please feel free to let us know.

---

> > ### Comment · Reviewer_1xqQ · 2025-04-03
> >
> > Thanks for the detailed reply. But I do not think my concern in Q3 has been addressed yet. I asked the authors how initialization methods affect the **results** and the **gradients**. The authors give the results under Xavier initialization and claim they will provide the gradient results later.
> >
> > Actually, I mentioned **four initialization methods** in Q3---He initialization (with variance 2/d), LeCun initialization (with variance 1/d), the special cases 10/d and 1/10d. But **none of them** appears in the rebuttal. The authors only provided the results of Xavier initialization, I do not think the only result is enough to answer my question. I think discussing initialization is important, because a smaller weight initialization in networks with residual connections may relieve gradient exploding (**even there is normalization**), then a larger learning rate can be applied. I conducted related experiments on ResNet in early time so I am confident that this question is important. This is also why I mentioned the case "1/10d" in Q3.
> >
> > Therefore, I will decrease my score to 2 temporarily.

---

> > > ### Author Response · Authors · 2025-04-04
> > >
> > > **Response to Reviewer 1xqQ**
> > >
> > > Thank you for your detailed comments and for emphasizing the importance of weight initialization. In response to your suggestion, we conduct additional experiments examining four distinct initialization methods—He (with variance 2/d), LeCun (with variance 1/d), and the extreme variants 10/d and 1/(10d)—beyond the standard Xavier initialization previously reported. We provide our findings below and hope they address your concerns regarding how initialization schemes might influence training stability and final performance.
> > >
> > > ---
> > > ### **Weight Initialization Experiments**
> > >
> > > - **Experimental Setup** : We pre-train 400M-parameter Transformers on 30B tokens, using the controlled training random seed and hyperparameters (Section 4.1 in our paper), varying only the initialization method. All evaluations were performed under the same settings used in Table 2 of the paper.
> > >
> > > - **Table:**
> > > | 400M  || He (2/d) | LeCun (1/d) | 10/d   | 1/(10d) | Paper Baseline|
> > > |:-:|:-:|:-:|:-:|:-:|:-:|:-:|
> > > |Loss|Pre-LN|2.965|3.005|4.526|3.012|3.035
> > > ||Peri-LN|2.929|2.915|3.027|2.902|2.916
> > >
> > > - **Figures** : [Link to Figures](https://anonymous.4open.science/r/ICML2025_Peri_LN-F359/README.md)
> > > We extend our paper by conducting an analysis of four distinct initialization methods—He (2/d), LeCun (1/d), and two extreme variants (10/d and 1/(10d)). The figures present the following:
> > >
> > >     - Training loss curve
> > >     - Gradient norm curve
> > >     - Forward growth patterns of hidden states magnitude and variance at the final stage
> > >     - Backward gradient norm and variance at the final stage
> > >
> > > ---
> > >
> > > ### **Discussion**
> > > 1. **Forward Growth Patterns of Hidden States**
> > > Visualizing the forward-pass hidden-state variances confirms that Pre-LN exhibits exponential-like growth in intermediate activations, particularly under high variance (10/d). By contrast, Peri-LN “self-regulates” these activations more effectively, staying closer to a stable range throughout training.
> > > To investigate whether using a smaller weight initialization in networks with residual connections can help mitigate explosion, we would like to highlight the smallest variance, $1/(10d)$ results. As shown in the figure of "Forward growth patterns" Section, this setting still exhibits large hidden-state magnitudes and variance in the forward path, suggesting that simply reducing the weight initialization may not be sufficient.
> > >
> > > 2. **Comparison of Training Loss**
> > >    Under all tested initialization conditions, Peri-LN consistently converges to lower training loss compared to Pre-LN. Even when we vary the weight initialization variance substantially (from 1/(10d) to 10/d), Peri-LN maintains an advantage in final loss.
> > >
> > > 3. **Early-Stage Instability in Pre-LN**
> > >    As also noted in our original paper, we observed that Pre-LN often exhibits pronounced spikes in both loss and gradient norm during the early training stages, especially for larger variances (e.g., 10/d). These spikes are less severe or absent under Peri-LN.
> > >
> > > 4. **Sensitivity to Weight Initialization Variance**
> > >    Pre-LN shows greater sensitivity to different initialization distributions, leading to more variation in final outcomes. This aligns well with our earlier observations in Table 2 of the paper, where Pre-LN typically underperforms or diverges for certain initialization settings (notably 10/d). In our tests, Pre-LN worked best with He initialization, while Peri-LN is robust across a broader range of settings.
> > >
> > > 5. **Divergence for Large Variance**
> > >    Notably, under the 10/d initialization, Pre-LN diverges almost immediately, whereas Peri-LN remains stable. This suggests that Peri-LN may offer a safeguard against the runaway activations that arise under large-variance conditions in deep residual networks.
> > >
> > > 6. **Backward Gradient Norm and Variance at the Final Stage**
> > > Finally, analyzing the gradient norm at later training stages reveals no substantial change from our earlier conclusion.
> > >
> > > ---
> > > ### **Final Remarks**
> > >
> > > We hope these additional experiments demonstrate that our core findings regarding Peri-LN’s stability and robustness hold across a wide spectrum of initialization choices—from conventional (He, LeCun) to more extreme settings (10/d, 1/(10d)). We appreciate your suggestion to explore these variants, as they further highlight Peri-LN’s advantages in curbing gradient and activation spikes, even under challenging initialization conditions.
> > >
> > > If there are any additional experiments or questions you would like us to pursue, we would be glad to discuss them. We plan to include these new initialization results in the revised manuscript to reinforce our claim that Peri-LN’s benefits persist under varied starting points. Thank you again for your thoughtful feedback, which has greatly helped us strengthen the paper.

---

### Official Review · Reviewer_nnDw · 2025-03-14

**Overall Recommendation:** 2

**Summary:**

This paper investigates the effectiveness of  position where layer normalization (LN) (mainly its reduced version RMSNorm)  is placed in the Transformer architecture.  It also provides analyses from the perspective of activation/gradient propagation in the network to explain why a position LN  placed usually works better. In particular, it claims that the previous Pre-LN and Post-LN architecture are prone to vanishing gradients and "massive activations". It advocates to place LN peripherally around sublayer, termed Peri-LN (There exists similar usage of LN in the Transformer from previous work, as pointed in the paper). The experiments are conducted on Transformers up to 3.2B parameters, showing that Peri-LN achieves more balanced variance growth, steadier gradient flow, and convergence stability.


## update after rebuttal:

I have read the responses and other reviewers comments. My concerns on claims and Proposition 3.1 still hold. The two main claims of this paper:  1. Pre-LN  has exploding gradient (Proposition 3.1. (1);  2. Post-LN has  vanishing gradient are not well validated by experiments.  E.g., this paper should provide the results (gradients) to support the claims under different weight initialization and weight decay (if the theory and analyses hold, it should hold under different weight initialization and weight decay).  However, this paper does not provide the results for Post-LN even in rebuttal and  the provided results for the Pre-LN are also not convincing (i.e., I donot find exploding gradient).

I keep may score, and towards reject this paper.

**Claims And Evidence:**

The main claims of this paper is that: (1)  it claims that the previous Pre-LN and Post-LN architecture are prone to vanishing gradients and "massive activations"; (2)  Peri-LN achieves more balanced variance growth, steadier gradient flow, and convergence stability.  I believe claims 2 are mostly correct and convincing, based on the experiments and my understanding. However,  I am not convinced by the claim 1.  Even though this paper provides informal (the so-called) theory and experiments to support claim 1, the experiments are not sufficient to support, e.g., it does not consider the affects of weight's variance and the optimizer, please see the comments in the experiments design, and I also have concerns on theory (see the comments in the theoretical claims).



Besides, some description is somewhat over-claimed, e..g, "we provide fresh insights into where and when it may offer advantages over more widely adopted LN placements." .  I believe the analyses on the position of normalization using activation/gradient propagation are widely used in previous work (e..g, the paper introduces Pre-LN architecture), please see the survey paper [1] for details.



Ref:  [1] *Normalization Techniques in Training DNNs: Methodology, Analysis and Application*, TPAMI 2023.

**Essential References Not Discussed:**

I believe this paper provides overall references

**Experimental Designs Or Analyses:**

Even thought the experiments show a results supporting the claim 1 (that the previous Pre-LN and Post-LN architecture are prone to vanishing gradients and "massive activations") , I still have concerns on the experimental setups. My concerns are follows:

(1) The activation and gradients is related to the variance of weight matrix initially, this paper should provide the details of initialization of  weight matrix, and further investigates whether  claim 1 holds when varying the  variance of weight matrix initially.  It is clear that normalization has the scale invariant property during forward process, but has the inverse scale property during back-propagation,  and the scale of weight matrix apparently affects the results.  Besides, this paper uses weight decay in the experiments,  I think this paper should further consider whether the claim 1 hold if weight decay removed or consistently hold if weight decay varied.

(2) The analyses in theory is based on the gradients (e.g., SGD), not provide the analyses when using Adam optimizer. However, the experiments conducted in this paper only use the adam optimizer, not using SGD. It is not sufficient to support the theory.

**Methods And Evaluation Criteria:**

The proposed method and evaluation criteria is overall make sense. But ithe experiments are not sufficient to support, e.g., it does not consider the affects of weight's variance and the optimizer, please see the comments in the experiments design.

**Other Comments Or Suggestions:**

The experiments are mainly conducted on RMSNorm, this paper should provide the background of Layer Normalization and RMSNorm, in case of the reader is not familiar to them.

**Other Strengths And Weaknesses:**

NA

**Questions For Authors:**

See the questions in Theoretical Claims (comments)

**Relation To Broader Scientific Literature:**

This paper provides clear routing how this paper builds on.

**Theoretical Claims:**

The main (informal) theoretical claims is Proposition 3.1.

Indeed, I have concerns on this Proposition:

(1) why this paper only consider  the gradient of $W^{(2)}$, why not consider the gradient of   $W^{(1)}$ in the MLP, and further other weights in Self-Attention?



(2)  The vanishing gradient of Post-LN is base on the description that "when a massive activation $\|h\|$ occurs, Norm() introduces an overly suppressing factor $\|x+a\|$", why is that,  note that the bound is relating to $\frac{\|h\|}{\|x+a\|}$, when this paper assume $\|h\|$ is also massive.

(3) The exploding gradient of Pre-LN is based on the description that "when a massive activation $\|h\|$"occurs,  $\|\frac{\partial{L}}{\partial{W^{(2)}}}\|$ can arise? why $\|h\|$ definitely  occurs? even that, why leading to training instability? especially, the optimizer is Adam used in this paper, which can well remove out the scale of gradients.

---

> ### Author Rebuttal · Authors · 2025-03-31
>
> ### **1) Considering $W^{(2)}$ in the MLP**
> Following Theorem 1 in [1], we analyze the last layer, as the gradient norm at the final layer is empirically known to be the most unstable (see Figure 1 in the paper). We choose $W^{(2)}$ (the final linear projection in the MLP) as a representative example because it most directly feeds into the residual connection $x+a$, making it clearer to illustrate how gradient norms can explode or vanish.
>
> [1] Xiong et al. "On layer normalization in the transformer architecture." ICML2020.
>
> ---
> ### **2) Post-LN and Vanishing Gradients**
> In Proposition 3.1, the bound $\frac{\||h\||}{\||x+a\||}$ involves not only $\||h\||$ but also $\||x+a\||$. Since $h$, $x$, and $a$ are interrelated, what ultimately matters is their relative magnitudes. The theory indicates that the Post-LN structure introduces $\||x+a\||$ into the bound, whereas Peri-LN introduces only $\||a\||$. Experimentally, we confirm that Post-LN exhibits vanishing gradients (see Figure 6(b, d) and [2]). As shown in Appendix D, the presence of a normalization layer significantly influences the gradient scale not only in Peri-LN but also in Post-LN—an observation consistent with [1] and [2]. For more discussion, we kindly refer to Response #2 of *Reviewer AtH3*.
>
> [2] Kedia et al. "Transformers get stable: an end-to-end signal propagation theory for language models." ICML2024.
> [3] Fishman et al. "Scaling FP8 training to trillion-token LLMs." ICLR2025.
>
> ---
> ### **3) Pre-LN and Exploding Gradients**
> As discussed in our paper (Figures 1 and 6) and by Sun et al. [4], we observe that Pre-LN architectures can produce large activation values. Fishman et al. [3] and Wortsman et al. [5] note that quadratic computations in both the Attention [5] and MLP [3] modules can yield large activations. In Pre-LN architectures, these outputs are not regulated by normalization at the sub-layer level. Consequently, high-variance spikes in intermediate activations can lead to training instabilities because, in Proposition 3.1, $\||h\||$ appears in the numerator of the gradient bound, posing a risk of gradient explosion. Although Adam adaptively rescales gradients, raw gradient spikes may still destabilize updates or force Adam to make abrupt learning rate adjustments (see Section C.1 in the supplementary, where Post-LN uses a lower LR). Indeed, as our paper shows, Pre-LN experiences gradient spikes and training instability despite Adam’s adaptive rescaling. Our main point is that Peri-LN mitigates these large activations at the sub-layer output, reducing the risk of high-magnitude raw gradients before the optimizer’s moment-based rescaling takes effect.
>
> [4] Sun et al. "Massive activations in large language models." COLM 2024.
> [5] Wortsman et al. "Small-scale proxies for large-scale transformer training instabilities." ICLR 2024.
>
> ---
> ### **4) Weight Initialization & Decay**
> - **Weight Initialization**: Due to limited space, we kindly refer you to Response #3 of *Reviewer 1xqQ*.
> - **Weight Decay**: We conduct additional studies for various weight decay condition for both Pre-LN and Peri-LN architectures.  As shown in the table, Peri-LN continues to offer better performance than Pre-LN under the same settings. We will include this detailed ablation studies in the revised manuscript to solidify that our findings (especially large hidden state variance) still hold under varied weight decay and initialization methods. Experimental settings are in Response #3 of *Reviewer 1xqQ*.
> |400M||Decay=0|0.0033|0.033|0.33|
> |-|-|-|-|-|-|
> |Loss|Pre-LN|3.03|3.03|3.03|3|
> ||Peri-LN|2.94|2.94|2.93|2.90|
> |Avg.|Pre-LN|49.26|49.18|49.01|49.51|
> ||Peri-LN|51.41|51.14|50.68|52.13|
> ---
> ### **5) Theory references SGD, but experiments use Adam**
> The key difference between Adam and SGD lies not in the gradients themselves but in how the learning rates are adjusted afterwards—Adam employs adaptive learning rates, while SGD uses a fixed one. Our theoretical analysis focuses on the structural characteristics of the raw gradients rather than assuming any specific optimizer behavior. Therefore, theoretical analyses on the gradients themselves remain valid regardless of whether SGD or Adam is used in the experiments.
>
> ---
> ### **6) Over-claimed novelty & Provide the background of LN**
> Since prior work has analyzed gradient behaviors of Post-&Pre-LN, our central contribution is to consolidate and extend these insights to a third placement—Peri-LN—that recent models (e.g., Gemma2 & 3, Olmo2) employ with limited understanding. We will moderate the overall expressions and provide a concise background on normalization layers.
>
> ---
> ### **Concluding Remarks**
> Your feedback has been instrumental in this process, and we sincerely extend our gratitude for your invaluable insights. Should you have any inquiries or require clarifications about our rebuttal, please don't hesitate to reach out. We are eager to address any concerns and elucidate potential ambiguities in greater depth.

---

> > ### Comment · Reviewer_nnDw · 2025-04-07
> >
> > Thanks for the response to my comments.  I still have concerns on the claims that the previous Pre-LN and Post-LN architecture are prone to vanishing gradients and "massive activations".  The authors do not directly respond my concerns on "(1) The activation and gradients is related to the variance of weight matrix initially, this paper should provide the details of initialization of weight matrix, and further investigates whether claim 1 holds when varying the variance of weight matrix initially." . This paper only provide the final results (e.g, loss/ avg. which is not important, since that the prei-LN has been proposed in previous papers), but I care is whether the claims that "the previous Pre-LN and Post-LN architecture are prone to vanishing gradients and "massive activations"" still hold under different weight initialization and weight decay.  The authors should provide the gradients and other evidence to support the claims.
> >
> >
> >
> > Besides,  I still have concerns on Proposition 3.1. The authors reply that "we analyze the last layer, as the gradient norm at the final layer **is empirically known to be the most unstable** (see Figure 1 in the paper). We choose W(2) (the final linear projection in the MLP) as a representative example because it most directly feeds into the residual connection x+a, making it clearer to illustrate how gradient norms can explode or vanish.". It seems the theory is based on empirical observation?  In another word, why W(1) is stable? or the gradient of W(1) has no affects on the overall gradients? I think this paper should pay more attention to clarify it.
> >
> >
> >
> > As to the "**Theory references SGD, but experiments use Adam**", why not attempt to train the model using SGD, if the Peri-LN has the  so-called stable gradients?

---

> > > ### Author Response · Authors · 2025-04-09
> > >
> > > ### **1. “the claims still hold under different weight initialization”**
> > > In prior works [1, 2], many findings on Post-LN and Pre-LN were discussed. Since prior works [1,2] mainly focused on the initialization phase, the primary claim—that Pre-LN can exhibit large activation variance—and the behavior of Peri-LN have not been thoroughly investigated. This gap persists even when considering final loss behaviors. To address this gap, we conducted additional experiments and analyses across four distinct initialization methods—He ($\tfrac{2}{d}$), LeCun ($\tfrac{1}{d}$), and the more extreme variants $\tfrac{10}{d}$ and $\tfrac{1}{10d}$. We use the same settings outlined in Section 4.
> > >
> > > - **Table:**
> > > |400M||He (2/d)|LeCun (1/d)|10/d|1/(10d)|Paper|
> > > |:-:|:-:|:-:|:-:|:-:|:-:|:-:|
> > > |Loss|Pre-LN|2.965|3.005|4.526|3.012|3.035
> > > ||Peri-LN|2.929|2.915|3.027|2.902|2.916
> > >
> > > - **Figures** : [Link to Figures - Weight Init](https://anonymous.4open.science/r/ICML2025_Peri_LN-F359/README.md)
> > > We include:
> > >     - Training loss curve
> > >     - Gradient norm curve
> > >     - Forward growth patterns of hidden states magnitude and variance at the final stage
> > >     - Backward gradient norm at the final stage
> > >
> > > Visualizing the forward-pass hidden-state variances confirms that Pre-LN exhibits exponential-like growth in intermediate activations. By contrast, Peri-LN regulates these activations more effectively, staying closer to a stable range throughout training.
> > >
> > > ---
> > > ### **2. “.. different weight decay.”**
> > > Due to the limited time and resources, we couldn’t finalize analysis on weight decay experiments. Instead, we provide training loss and gradient-norm curves under varying weight decay:
> > >
> > > - **Figures** : [Link to Figures - Weight Decay](https://anonymous.4open.science/r/ICML2025_Peri_LN_decay-7DDD/README.md)
> > >
> > > As noted in our paper, we observed that Pre-LN often exhibits pronounced spikes in both loss and gradient norm during the early training stages. These spikes are less severe or absent under Peri-LN.
> > >
> > > ---
> > > ### **3. "It seems the theory is based on empirical observation? In another word, why W(1) is stable?”**
> > > We would like to clarify that the theoretical analysis is not based on empirical observations. Rather, our decision on which component of the model to analyze is informed by empirical observations.
> > >
> > > Also, we did not intend to suggest that $W^{(1)}$ is stable. Rather, we focus our analysis on $W^{(2)}$ because it empirically exhibits the most severe gradient instability. Consequently, Section 3.4 focuses the theoretical lens to this final MLP layer $W^{(2)}$. This choice is motivated by the desire to give a mathematically rigorous and tractable argument at the point in the network where instability is most critical.
> > >
> > > Previous studies have primarily examined the initialization phase only [1, 2] and likewise home in on the final layer for tractability [1]. Our work extends beyond initialization, uncovering how different placements of layer normalization can trigger or mitigate instabilities throughout training.
> > >
> > > We would like to emphasize that we pair our theory with extensive experiments to illustrate how Transformers behave differently according to the placement of layer normalization in practice.
> > >
> > > ---
> > > ### **4. “why not attempt to train the model using SGD, if the Peri-LN has the so-called stable gradients?”**
> > > In line with our previous discussions, we emphasize that our theory is not dependent on the choice of optimizer. Rather, it relies on the hidden states and gradient scales of the final hidden layers, as described in the paper.
> > >
> > > Regarding the use of SGD, using SGD for training Transformers is not a common practice. As Zhang et al. [3] point out, Transformer-based models tend to perform worse with SGD than with Adam by a considerable margin. One reason is that SGD struggles to handle the heterogeneity across different blocks. Although these aspects are certainly intriguing and warrant further investigation, they lie beyond the scope of our current work, as Zhang et al. also note.
> > >
> > > Nonetheless, we conducted additional experiments using SGD, as recommended by the reviewer. We are searching for U-shaped patterns during the learning rate exploration for both Pre-LN & Peri-LN as shown in the figure titled "Learning Rate Exploration." We observed that: (1) SGD performs worse than Adam, consistent with findings reported in [3]; and (2) Peri-LN demonstrates better performance than Pre-LN. Please refer to the link below.
> > > - [Link to Figures - SGD](https://anonymous.4open.science/r/ICML2025_Peri_LN_SGD-EA17/README.md)
> > >
> > > ---
> > > ### **Remarks**
> > > We hope these additional experiments and clarifications resolve your concerns. We deeply appreciate your guidance throughout this process.
> > >
> > > ---
> > > ### **Reference**
> > > [1] Xiong et al. "On layer normalization in the transformer ..." ICML 2020.
> > > [2] Kedia et al. "Transformers get stable: an end-to-end signal propagation ..." ICML 2024.
> > > [3] Zhang et al. "Why Transformers Need adam: A hessian perspective." NeurIPs 2024.

---

### Official Review · Reviewer_AtH3 · 2025-03-14

**Overall Recommendation:** 5

**Summary:**

This paper examines a layernorm "layout" in the transformer architecture called PeriLN. The PeriLN combines Prelayernorm with a module out layernorm (similar to post layernorm but before the residual stream). The authors provide intuition for how this addresses weaknesses in the Post layernorm and prelayernorm layouts along with supporting theoretical statements. The authors provide comprehensive experiments showing that this layout dominates the other layouts in performance and stability on LLM experiments.

**Claims And Evidence:**

Yes the claims are supported by clear and convincing evidence.

**Essential References Not Discussed:**

None.

**Experimental Designs Or Analyses:**

Yes the designs are sound.

**Methods And Evaluation Criteria:**

Yes the methods and evaluation are solid.

**Other Comments Or Suggestions:**

I think QV norm discussion could be moved into the main paper because it is quite interesting and important in the "layernorm in transformers" design space.

**Other Strengths And Weaknesses:**

The paper is written very clearly and the experiments are very solid, especially the learning rate sweeps (which I believe use muP).

**Questions For Authors:**

In olmo2 there is only a module output layernorm and not the prelayernorm? So this differs from PeriLN?

Does the finding about QK normalization say that it is not needed with PeriLN?

It seems that pre + post layernorm should be worse than periLN, perhaps that can be confirmed experimentally?

My takeaway from Xiong et al. [1] was that Postlayernorm will lead to gradient norm blowup, but from this paper it seems like the issue is actually gradient norm vanishing. How should I reconcile this?


[1]  On Layer Normalization in the Transformer Architecture - Ruibin Xiong, Yunchang Yang, Di He, Kai Zheng, Shuxin Zheng, Chen Xing, Huishuai Zhang, Yanyan Lan, Liwei Wang, Tieyan Liu

**Relation To Broader Scientific Literature:**

This paper contributes to the literature on understanding training dynamics and layernorm in architectures. It identifies the weaknesses and strengths of post and prelayernorm, providing substantial evidence for a better alternative.

**Theoretical Claims:**

I did not check correctness explicitly but they seem reasonable.

---

> ### Author Rebuttal · Authors · 2025-03-31
>
> ### **1) Confirming That Pre + Post LayerNorm is Worse Than Peri-LN**
> >*“It seems that pre + post layernorm should be worse than periLN, perhaps that can be confirmed experimentally?”*
>
> In response to the reviewer's comment, we additionally conduct further experiments on LN placements to compare different combinations (referred to as A, B, and C positions in Figure 2 of the paper). We add configurations where LN is placed at both A + C (akin to combining Pre- and Post-LN), as well as only at B, to compare them with Peri-LN at final training loss under the controlled same training seed. We pre-train the 400M-parameter Transformers on 30B tokens each, using the same training configurations described in the paper. As aligned with Xiong et al. [1], our new results confirm that placing LN exclusively at C leads to training instability or suboptimal performance. In particular, the A + C configuration inherits characteristics of Post-LN (large gradient norm shifts), forcing the use of smaller learning rates and still resulting in lower overall performance than Peri-LN architecture. We will include additional learning rate sweep results and detailed training loss curves for the additional A+C and B experiments in the revised manuscript to more comprehensively illustrate these differences.
>
> |400M|A + C|Post-LN|B|Peri-LN|
> |:-:|:-:|:-:|:-:|:-:|
> |Loss|3.01|3.05|Diverged|2.91|
>
> [1] Xiong, Ruibin, et al. "On layer normalization in the transformer architecture." ICML 2020.
>
> ---
> ### **2) Reconciling Gradients Blowup vs. Vanishing Gradients in Post-LN**
> >*“My takeaway from Xiong et al. [1] was that Postlayernorm will lead to gradient norm blowup, but from this paper it seems like the issue is actually gradient norm vanishing. How should I reconcile this?”*
>
> Thank you for raising this point. Our layer-wise observations in Post-LN (Figure 6 in the paper) indeed show signs of gradient vanishing through layers, yet we also observe strong gradient spikes (i.e., blowups in the total gradient summation) at various stages of training. This aligns with Xiong et al. [1], where the large shift in Post-LN gradients causes instabilities that lead to sudden spikes. In essence, both Xiong et al. and our Proposition 3.1 suggest that the gradient scale in Post-LN can swing dramatically. When examining training iteratively (step by step), we observe occasional gradient spikes (blowups). However, across the broader span of training, Proposition 3.1 and [2] show that Post-LN gradients ultimately exhibit a vanishing tendency overall—consistent with Figure 6.
> We appreciate the chance to clarify further. In the revised manuscript’s supplementary material, we will include additional plots demonstrating both the micro-scale spikes in both gradient and loss over the course of training.
>
> [2] Kedia, Akhil, et al. "Transformers get stable: an end-to-end signal propagation theory for language models." ICML 2024.
>
> ---
> ### **3) QK-Norm Discussion**
>
> We agree that QK-Norm plays an increasingly important role in modern Transformers. As you suggest, we will move or more prominently feature the QK-Norm discussion from the supplementary section into the main paper.
>
> ---
> ### **4) Clarifying Olmo2 vs. Peri-LN**
>
> Yes, as noted in Appendix G, the Olmo2 architecture slightly differs from Peri-LN. Peri-LN uses both a Pre-LN and an Output-LN, whereas Olmo2 relies on QK-Norm plus the output LN (No Pre-LN). From our experiments, applying only an output LN (or only a QK-Norm) proved insufficient to stabilize training under challenging hyperparameter settings, which is consistent with remarks in the Olmo2 paper [3].
>
> [3] OLMo, Team, et al. "2 OLMo 2 Furious." arxiv 2024.
>
> ---
> ### **5) Is QK-Norm Unnecessary for Peri-LN?**
>
> While Peri-LN alone provides robust training dynamics, QK-Norm can still enhance performance. In response to reviewers comment, we conducted additional experiments that confirm combining Peri-LN with QK-Norm yields slight improvements in training loss. We pre-train the 1.5B-parameter Transformers on 30B tokens each, using the same training configurations described in the paper. This observation is consistent with prior work [4] indicating that QK-Norm can synergize with various LN placements. We will include these new results in the revised manuscript with detail, along with references to recent works like gemma3 [5], which successfully integrate Peri-LN and QK-Norm.
> | 1.5B | Peri-LN | + QK-Norm |
> |--|---|-|
> |Loss|2.722|2.711|
>
> [4] Wortsman, Mitchell, et al. "Small-scale proxies for large-scale transformer training instabilities." ICLR 2024.
> [5] Team, Gemma, et al. "Gemma 3 Technical Report." arXiv 2025.
>
> ---
> ### **Concluding Remarks**
> We believe our additional experiments and clarifications—regarding LN placements, QK-Norm, gradient behavior—will strengthen the paper significantly. Your insightful guidance has been instrumental in refining our analysis. We will incorporate your valuable comments into the revised manuscript.

---

### Decision · Program_Chairs · 2025-05-01

**Decision:**

Accept (poster)

**Comment:**

This paper provides an analysis of different layer normalization strategies and how they impact the training dynamics of large-scale transformers, finding that peripherally bracketing normalization layers around submodules (Peri-LN) can improve stability relative to the pre- and post-LN baselines.

This paper received mixed reviews, with some reviewers finding strength in the comprehensive empirical analysis and others questioning claims about exploding/vanishing gradients and highlighting gaps between theory and practice. Nevertheless, there is wide agreement that  the paper does provide convincing evidence that Peri-LN achieves more stability during training. Given the breadth and depth of empirical investigation, it is not necessary for the theoretical analysis to provide a rigorous explanation for all the observed phenomena and can instead simply serve to provide some useful guiding intuition to accompany the experiments. As such, I believe this paper will provide a valuable contribution to the ICML community and I recommend acceptance.